# Recent Advances in the Therapeutic Potential of Bioactive Molecules from Plants of Andean Origin

**DOI:** 10.3390/nu17111749

**Published:** 2025-05-22

**Authors:** Carlos Barba-Ostria, Jéssica Guamán-Bautista, Augusto A. Tosi-Vélez, Juan A. Puente-Pineda, Melanie A. Cedeño-Zambrano, Enrique Teran, Linda P. Guamán

**Affiliations:** 1Escuela de Medicina, Colegio de Ciencias de la Salud Quito, Universidad San Francisco de Quito USFQ, Quito 170901, Ecuador; cbarbao@usfq.edu.ec (C.B.-O.); eteran@usfq.edu.ec (E.T.); 2Instituto de Microbiología, Universidad San Francisco de Quito USFQ, Quito 170901, Ecuador; 3Facultad de Ciencias de la Hospitalidad, Universidad de Cuenca, Cuenca 010201, Ecuador; jessica.guaman@ucuenca.edu.ec (J.G.-B.); augusto.tosi@ucuenca.edu.ec (A.A.T.-V.); 4Centro de Investigación Biomédica (CENBIO), Facultad de Ciencias de la Salud Eugenio Espejo, Universidad UTE, Quito 170527, Ecuador; juan.puente@ute.edu.ec (J.A.P.-P.); melanie.cedenio@ute.edu.ec (M.A.C.-Z.)

**Keywords:** Andean plants, bioactive compounds, antioxidant, antimicrobial, *Chenopodium quinoa*, *Amaranth* spp., *Lupinus* spp.

## Abstract

**Background:** Andean plants are rich in bioactive compounds shaped by extreme environmental conditions, contributing to their antioxidant, antimicrobial, and anti-inflammatory properties. This review explores their phytochemical composition, biological activities, and therapeutic potential in modern medicine and nutrition of three plants of Andean origin. **Methods:** A literature review of peer-reviewed studies was conducted, focusing on key species such as quinoa (*Chenopodium quinoa*), amaranth (*Amaranthus* spp.), and lupin (*Lupinus* spp.), selected for this review due to their Andean origin, long-standing role in traditional diets, and growing scientific interest in their unique phytochemical profiles and therapeutic potential. This analysis covers their phytochemistry, bioactivities, and the influence of environmental factors on compound potency. **Results:** These Andean-origin plants contain flavonoids, terpenoids, alkaloids, and phenolic compounds that support antioxidant, antimicrobial, anti-inflammatory, and anticancer activities. High-altitude conditions enhance the biosynthesis of these bioactives, increasing their therapeutic value. Quinoa, amaranth, and lupin show strong potential for dietary and pharmaceutical applications, particularly in metabolic health and disease prevention. Additionally, preclinical studies and clinical trials have begun exploring the efficacy of these compounds in preventing and treating metabolic and chronic diseases. **Conclusions:** Andean plants are a valuable source of functional bioactive molecules with diverse health benefits. Future research should optimize cultivation strategies and explore novel applications in nutrition and medicine.

## 1. Introduction

In recent decades, there has been a remarkable resurgence of interest in functional foods, driven by a growing awareness of the relationship between diet and health [1]. These foods—both naturally rich in bioactive compounds and fortified with beneficial ingredients—offer advantages beyond basic nutrition by contributing to disease prevention and overall well-being, helping mitigate the risk of chronic disease risks and promote a better quality of life. For instance, antioxidants play a crucial role in neutralizing superoxide radicals, thereby reducing the risk of non-communicable diseases such as cancer, diabetes, cardiovascular, neurological, kidney, and liver conditions [2]. In parallel, natural antimicrobials, including secondary metabolites produced by plants, animals, and microorganisms, inhibit pathogenic microorganisms, thus enhancing food safety, extending shelf life, and preventing foodborne illnesses [3].

Furthermore, there is a growing interest in bioactive molecules from natural sources, particularly plants of Andean origin, which are known for their exceptional phytochemical composition and historical significance in traditional medicine and nutrition [1]. The extreme environmental conditions of the Andean region—characterized by high altitudes, intense ultraviolet radiation, and nutrient-poor soils—have driven the evolution of species such us quinoa, amaranth, and lupin in Bolivia and Peru with unique adaptive phenotypes, resulting in the accumulation of diverse secondary metabolites (e.g., phenolic compounds, flavonoids, terpenoids, and alkaloids) that confer antioxidant, antimicrobial, and anti-inflammatory properties [4,5].

Functional foods and nutraceuticals derived from these plants, quinoa—(*Chenopodium quinoa*), amaranth (*Amaranthus* spp.), and lupin (*Lupinus* spp.)—have attracted considerable attention for their nutritional density and bioactivity, positioning them as promising candidates for dietary supplements and pharmaceutical applications [6,7]. Numerous studies have demonstrated their antioxidant, antimicrobial, anticancer, and metabolic-regulating properties, underscoring the potential of these Andean species as valuable tools for enhancing human health and ensuring food security [8,9,10].

We conducted a comprehensive literature search on bioactive compounds from *Chenopodium quinoa*, *Amaranthus* spp., and *Lupinus* spp., including their biological activities and health benefits, using the scientific databases Web of Science (WoS) and Scopus. We selected peer-reviewed original research articles, review papers, conference proceedings, and book chapters published in English or Spanish within the last ten years. Exceptionally, we included a limited number of older publications when they provided fundamental information on the botanical or chemical characteristics of the studied plants.

This review provides a comprehensive analysis of the bioactive compounds present in Andean-origin plants, their biological activities, and their potential health benefits, thereby integrating insights from phytochemistry, ethnobotany, and biomedical research to highlight their therapeutic potential and applications in modern nutritional science and medicine.

## 2. Ancestral Edible Plants and Their Historical Significance

Ancestral edible plants have played a fundamental role in human nutrition, medicine, and agricultural practices for millennia. Defined as plants that have been traditionally cultivated and consumed by indigenous cultures, these species provide essential nutrients and bioactive compounds while preserving valuable traditional knowledge passed down through generations [11]. The study of these plants offers significant opportunities for the development of sustainable and health-promoting food products [12]. Additionally, the use of by-products, such as peels, seeds, and pulp, enhances resource efficiency by serving as concentrated sources of bioactive compounds with applications in food, medicine, and pharmacology [13]. By integrating these elements into modern diets, societies can improve nutritional quality while reducing food waste and promoting sustainability [14].

### 2.1. The Andean Region as a Center of Plant Diversity

The Andean region, also known as Highlands in some textbooks, has been a crucial center for plant diversification due to its unique geological history and environmental conditions. The uplift of the Andes created a variety of altitudes and climates, leading to the evolution of a rich tapestry of flora that has sustained human societies for millennia. The region’s biodiversity has not only influenced ecological dynamics, but has also shaped agricultural practices and cultural traditions [15].

Also, Andean cultures have historically maintained deep connections to their natural landscape, with food playing a significant role in cultural and religious traditions. Food preparation is often viewed as a sacred act linked to ethnic identity, and communal feasting continues to reinforce social bonds. The Spanish conquest introduced foreign plants and animals, reshaping Andean foodways by blending indigenous and European culinary traditions. An interdisciplinary perspective, integrating anthropology and art history, reveals the complexity of Andean food systems which continue to influence regional cuisines today. The resilience and adaptability of Andean agricultural heritage are reflected in the persistence of traditional crops such as quinoa, amaranth, and lupinus [16].

### 2.2. Quinoa: The “Mother Grain” of the Andes

Chenopodium quinoa Willd., commonly known as quinoa, is an ancient grain native to the Andean highlands, with archeological evidence indicating its domestication around Lake Titicaca—shared by modern-day Peru and Bolivia—as early as 3000–5000 BCE. Revered by the Incas as the “Mother Grain”, quinoa played a central role in Andean agriculture and diets from the 12th to 15th centuries. Indigenous communities such as the Quechua and Aymara have long relied on quinoa not only as a nutrient-rich food source, but also as a component of cultural and medicinal practices, attributing digestive and immune-boosting properties to it [17]. Mainly associated with the High Andean plateau (Altiplano) at elevations around 3500 m.a.s.l., its cultivation spread to southern Chile during the Inca Empire. While possible secondary domestication events have been suggested in central Chile, the Andean Altiplano remains the most widely accepted center of origin [18].

Quinoa is distinguished by its wide array of bioactive molecules—phenolic compounds, fatty acids, peptides, and unique triterpenoids—which vary significantly with genotype, environmental conditions, and processing methods, shaping its unique biochemical and nutritional profile. With approximately 3000 varieties conserved in South American germplasm banks, quinoa has demonstrated remarkable adaptability to extreme conditions during over 5000–7000 years of Andean cultivation [19]. This review focuses on *Chenopodium quinoa* varieties native to the Andean region, where the crop originated and was subsequently disseminated across South America through ancient trade and livestock migration routes, becoming known by various local names [20].

In contemporary times, quinoa has gained global recognition as a superfood, leading to increased demand and economic opportunities for Andean farmers. This expansion has facilitated the preservation of traditional farming methods while providing sustainable income sources for local communities. The United Nations’ designation of 2013 as the “International Year of Quinoa” underscored its importance in global food security and nutrition [9].

Despite its modern commercialization, quinoa remains a staple in the Andean diet. In the districts of Quinoa and Acos Vinchos, it is commonly consumed in soups or as a side dish, with 58.3% of the population incorporating it into their daily meals. Most quinoa cultivation (66.7%) is for personal consumption, ensuring food security and reducing reliance on commercial grains. While traditional medicinal uses, such as its role as a purgative or colic remedy, have largely faded, quinoa’s resilience to harsh environmental conditions continues to make it a key crop for sustainable agriculture [21].

### 2.3. Amaranth: An Ancient Crop of the Americas

Amaranth (*Amaranthus* spp.) has been cultivated for thousands of years, particularly by ancient American civilizations such as the Mayans, Aztecs, and Incas. Highly valued for its nutritional properties, amaranth was consumed both as a vegetable and a grain, serving as a staple in pre-Columbian diets. The Aztecs considered it sacred, using it in religious ceremonies, which underscores its deep cultural significance.

Despite its historical importance, amaranth cultivation faced decline due to legislative restrictions imposed by colonial authorities, illustrating how external factors can impact traditional food systems. However, interest in amaranth has resurged due to its high protein content, essential amino acids, and adaptability to diverse climates. Today, amaranth remains an important crop in regions across Central and South America, Africa, and Asia. In West Africa, it is primarily used as a leafy vegetable, although its seeds are sometimes underutilized [22].

Amaranth’s adaptability has made it a promising crop for modern agriculture, particularly in regions affected by climate change. Its introduction to Ukraine between 1989 and 1992 highlights a growing recognition of its potential for food security and sustainability [23]. Research and agricultural initiatives, such as the Ecuadorian’s Amaranth Improvement Program for Cotopaxi (PROMAC), focus on selecting high-nutritional-value varieties suited to Andean conditions, further supporting agricultural resilience and sustainability.

### 2.4. Lupinus: A Versatile Andean Legume

Lupine seeds (*Lupinus mutabilis*), commonly known as tarwi, have also played a crucial role in Andean agriculture and nutrition for centuries. Indigenous Andean societies have long recognized their adaptability to high-altitude environments and poor soils. Rich in protein, fiber, and bioactive compounds, lupine seeds have been used both as food and in traditional medicine for managing conditions such as diabetes and hypertension [24].

Lupines have also been essential in sustainable agricultural practices due to their ability to fix atmospheric nitrogen, improving soil fertility and enabling crop rotation. This characteristic has historically made them valuable for maintaining productive farmlands in the Andean region [25].

Despite their long history of cultivation, lupine production has declined in recent years, particularly in Europe and Australia, raising concerns about the future role of this crop in food systems. However, scientific research has renewed interest in lupine seeds, emphasizing their potential health benefits in managing metabolic conditions and supporting sustainable agriculture [26]. Given their resilience to harsh environmental conditions and high nutritional value, lupines remain a promising candidate for addressing food security challenges in the Andes and beyond.

The Andean region has served as a cradle for diverse edible plants, shaping agricultural practices and cultural traditions for millennia. Quinoa, amaranth, and lupinus exemplify the deep historical, nutritional, and economic significance of ancestral plants. Their ability to withstand extreme environmental conditions, coupled with their exceptional nutritional profiles, positions them as critical crops for sustainable agriculture and food security in both local and global contexts. As modern research continues to explore their bioactive compounds and health benefits, these ancestral plants reaffirm their role as valuable resources for the future of food and nutrition.

## 3. The Andean Environment and Its Impact on Phytochemistry

The Andean ecosystem, shaped by extreme environmental conditions such as high altitudes (from 2500 to over 4000 m above sea level), intense ultraviolet (UV) radiation, and nutrient-poor soils, exerts profound selective pressures on plant physiology and phytochemistry [27]. These challenges have driven the evolution of distinct biochemical adaptations, allowing for Andean-origin plants to not only survive, but also produce bioactive compounds with significant therapeutic potential. Among these, *C. quinoa*, *Amaranthus hybridus*, and *L. mutabilis* stand out due to their remarkable resilience, nutritional density, and bioactive properties, making them essential crops for both local communities and global markets. Figure 1 illustrates the relationship between altitude, environmental stressors, and the types of phytochemicals produced by plants at various elevations.

### 3.1. Unique Ecological Factors of the Andes Region

The Andean environment fosters biochemical diversity by influencing nutrient uptake, secondary metabolite production, and stress-response mechanisms in plants. Quinoa and lupin, for instance, are highly adapted to these extreme conditions, thriving in nutrient-poor soils and enduring frequent droughts and frost events [4]. The mineral composition of Andean soils, particularly their high nitrate (NO_3_^−^) content, plays a crucial role in shaping plant metabolism. Quinoa ecotypes, such as Socaire, exhibit superior nitrate uptake efficiency, leading to enhanced protein and amino acid accumulation, which directly impacts both growth and nutritional quality [18].

Similarly, *L. mutabilis* has evolved specialized root systems that allow for it to fix atmospheric nitrogen, improving soil fertility and making it an integral part of traditional Andean crop rotation systems [28]. This ability not only enhances its own nutritional composition, but also benefits subsequent crops, ensuring sustainable agricultural practices. Meanwhile, *A. hybridus*, a species with deep roots in Andean agriculture, exhibits exceptional adaptability to soil variability and microclimatic fluctuations, leading to increased phytochemical accumulation that enhances its antioxidant and antimicrobial properties [29].

Beyond soil composition, altitude significantly impacts phytochemical production. Studies on Andean blueberries (*Vaccinium floribundum*) have shown that higher elevations correlate with increased polyphenol and anthocyanin accumulation, boosting antioxidant activity [30]. This pattern is also reflected in Figure 1, where higher altitudes are associated with the biosynthesis of complex antioxidant compounds such as anthocyanins, flavonols, and phenolic acids. A similar trend has been observed in quinoa, amaranth, and lupin, where high-altitude conditions enhance the biosynthesis of flavonoids, phenolic acids, and other protective secondary metabolites that contribute to their functional properties [9].

### 3.2. Adaptations of Andean Plants

To cope with the harsh Andean environment, *C. quinoa*, *A. hybridus*, and *L. mutabilis* have evolved an array of secondary metabolites—flavonoids, terpenoids, alkaloids, and phenolic compounds—that not only provide protection against abiotic stressors (e.g., UV radiation, drought), but also enhance their nutritional and therapeutic potential. These metabolic responses are essential for their resilience and survival, reflecting sophisticated biochemical flexibility [31].

Quinoa, for example, produces elevated levels of phenolic compounds when exposed to high UV radiation, which enhances its antioxidant properties [9]. Likewise, *L. mutabilis* exhibits significant phenotypic variation in response to soil type, with plants grown in silty loam soil demonstrating higher protein and isoflavone content compared to those in sandy clay loam, directly affecting their nutritional and medicinal value [32].

Amaranth, another crop deeply rooted in Andean agriculture, is known for its high adaptability to drought-prone environments. The extreme temperature fluctuations at high altitudes stimulate the accumulation of bioactive compounds such as betalains and phenolic acids, which contribute to its anti-inflammatory and cytoprotective effects [29].

In addition to their abiotic stress tolerance, Andean plants must balance their biochemical investment between pollination and herbivory defense. Research on *Haplopappus* spp. demonstrates how floral volatile profiles vary with altitude to optimize insect attraction while simultaneously deterring herbivores [33]. This ecological strategy is visually represented in Figure 1, where volatile and defensive compound production is associated with different elevation levels. Similarly, quinoa, amaranth, and lupin rely on these adaptive chemical strategies to maintain reproductive success and minimize biotic stress [34].

### 3.3. Influence of Environment on Bioactive Compound Potency

The extreme environmental pressures of the Andean region not only shape plant survival strategies, but also enhance their bioactive compound potency, making these crops valuable for human health. Studies indicate that quinoa grown at high altitudes accumulates higher levels of flavonoids and phenolic acids, contributing to its anti-inflammatory and antioxidant capacities [18]. Likewise, amaranth exhibits increased concentrations of bioactive peptides when cultivated under environmental stressors, enhancing its potential applications in functional foods and nutraceuticals [9]. Similarly, a study comparing quinoa varieties Regalona-Baer and Titicaca grown in different regions (Chile, Italy, Denmark) found that environmental conditions influenced their phytochemical profiles. Notable differences were observed in phenolic acids and flavonoids—especially vanillic acid, daidzein, and quercetin derivatives—while the total phenolic index was slightly lower in seeds grown in Italy compared to those from their original growing regions [35]. Also, a recent review selected studies from various research groups across different countries, providing a broad perspective on lupin samples and their quinolizidine alkaloid (QA) profiles. The research concluded that environmental conditions—such as light, temperature, soil pH, and drought stress—significantly affect QA synthesis, and that specific irrigation and nutrient regimes (e.g., low nitrogen, adequate potassium and phosphorus) help reduce QA accumulation [36].

*Lupin*, particularly *L. mutabilis*, has gained attention for its high protein content and unique phytochemical composition, which includes isoflavones and alkaloids with potential anticancer and lipid-lowering effects [32]. The Andean environment, with its fluctuating temperatures and nutrient-rich but arid soils, has been shown to drive variations in *L. mutabilis* seed composition, particularly in antioxidant capacity and β-tocopherol content [28].

As climate change continues to alter temperature, precipitation, and UV radiation levels, the synthesis of secondary metabolites in Andean plants is expected to intensify. Research suggests that elevated UV exposure leads to the increased production of protective compounds such as carotenoids and terpenoids, further enhancing their therapeutic potential [37]. Such shifts may also influence agricultural strategies, as traditional cultivation methods must adapt to maintain the bioactive integrity of these crops [34].

Furthermore, sustainable farming practices remain essential for preserving the biochemical diversity of Andean crops. Traditional Andean agricultural techniques, such as mixed cropping and rotational farming, play a crucial role in maintaining soil health and promoting phytochemical accumulation in quinoa, amaranth, and lupin [4]. The integration of indigenous knowledge with modern research provides a promising avenue for optimizing both yield and bioactive content in these crops.

The Andean ecosystem—defined by high altitudes, intense UV radiation, and nutrient-limited soils—has been instrumental in shaping the phytochemistry of *C. quinoa*, *A. hybridus*, and *L. mutabilis*. These crops have evolved distinct biochemical pathways that not only support their resilience to extreme conditions, but also enhance their nutritional and medicinal value. The synthesis of flavonoids, terpenoids, phenolic acids, and other bioactive compounds is a direct result of their adaptation to environmental pressures, positioning them as valuable sources of functional foods and therapeutic agents.

## 4. Bioactive Compounds in Quinoa (*Chenopodium quinoa*)

### 4.1. Phenolic Compounds

Phenolic compounds are among the most extensively studied bioactive molecules in quinoa due to their antioxidant properties and the significant variation in their presence across different quinoa cultivars. In a study examining 111 accessions of Chilean Altiplano quinoa (*Chenopodium quinoa* Willd) native to the Andean region of South America, the total phenolic content in seeds was found to vary widely, ranging from 35.51 mg to 93.23 mg per 100 g of seed dry weight. This substantial diversity in phenolic content highlights the biochemical variability among quinoa accessions, which could influence their functional properties. Notably, 72% of the total phenolic compounds were identified in the free phenolic fraction, making them more readily accessible for absorption, while the remaining 28% were bound phenolics that may be released during digestion, enhancing their potential benefits [38].

Within quinoa’s phenolic profile, several individual compounds are notable for their abundance and functional significance. Rutin and quercetin are two key flavonoids frequently identified in quinoa seeds, both known for their potent antioxidant activities. As shown in Figure 2, vanillic acid is another prominent phenolic acid found in quinoa [24]. Additionally, compounds like kaempferol, naringenin, and salicylic acid are especially prevalent in colored quinoa varieties, such as red and black quinoa, which generally have higher total phenolic and flavonoid content. For instance, black and red *C. quinoa* Willd has been documented with a phenolic content as high as 643.68 mg/100 g dry weight, indicating a phytochemical richness that surpasses that of white quinoa varieties [39].

Further research into colored quinoa has revealed a rich array of phenolic compounds, with over 430 polyphenols identified across various varieties. These polyphenols include phenolic acids, flavonoids, and flavonols, all contributing to the extensive phenolic profile of quinoa. A total of 67 polyphenols were identified as shared differential metabolites, indicating unique profiles across these cultivars. This compound diversity is visually represented in Figure 2, where different phenolics are illustrated according to quinoa color and variety. Black quinoa, in particular, exhibited the highest total phenolic content, while white quinoa had the highest flavonoid content, both of which are essential for their bioactive properties [39].

The phenolic composition of *C. quinoa* Willd (genotypes Besancon and Faro) is also subject to modification through processing techniques. Studies have shown that microwaving, germination, and other thermal treatments can increase the phenolic content and enhance the levels of bioactive compounds such as caffeic acid, rutin, and kaempferol. Germination is particularly effective, significantly raising the concentration of certain phenolic metabolites, including isoquercetin and 4-hydroxybenzaldehyde, especially during the third to fifth days of germination, with variability depending on the genotype of the quinoa used [40].

In addition to well-known compounds like rutin, quercetin, and vanillic acid, *C. quinoa* Willd grain extracts contain a variety of other phenolic compounds. Studies have identified catechin, caffeic acid, acacetin, and tangeretin in the methanolic extracts of quinoa. Acacetin and tangeretin, both flavonoids, further contribute to quinoa’s extensive phenolic profile [41].

Studies on related species, such as *Chenopodium pallidicaule* Aellen (kañihua) from Bolivia, have identified distinct phenolic profile. Both water-soluble and insoluble extracts contain resorcinol, 4-methylresorcinol, ferulic acid, kaempferol, and quercetin. The water-soluble extract uniquely includes catechin and vanillic acid, while catechin gallate is exclusive to the water-insoluble fraction [42].

Further studies on kañiwa from Peru have also demonstrated that this is a rich source of phenolic compounds, with the ‘Cupi’ and ‘Ramis’ varieties showing high levels of total phenolics (2.54 and 2.43 mg GAE/g, respectively). These values surpass those of grains like oats, buckwheat, quinoa, and rice [43].

Researchers have also identified unique flavonoids in *C. quinoa* Willd, including flavonols and flavanones, which contribute to its broad range of biological activities. The germination process enhances these compounds, leading to increased levels of free phenolic acids and bound flavonoids, thus amplifying the nutritional profile of quinoa [44].

In conclusion, quinoa’s phenolic compounds, including rutin, quercetin, vanillic acid, catechin, and caffeic acid, provide a robust bioactive profile that varies significantly across cultivars and can be further optimized using processing methods such as germination. This diversity underscores quinoa’s potential as a functional food ingredient and highlights its adaptability in delivering enhanced phenolic content through controlled processing techniques. The high variability in phenolic composition, both among quinoa varieties and due to processing, positions quinoa as a rich source of bioactive compounds with significant potential for health-promoting applications.

### 4.2. Polysaccharides

*C. quinoa* Willd seeds are a rich source of diverse polysaccharides, demonstrating significant nutritional potential and versatility in food and pharmaceutical applications due to their high sugar content (52.82–67.15%) and unique structural features. Notably, quinoa polysaccharides exhibit a triple-helix structure and semi-crystalline nature, with their thermal stability being observed through thermogravimetric analysis [45].

The primary crude polysaccharides extracted from *C. quinoa* Willd, referred to as Quinoa Crude Polysaccharides (QPS), include a soluble non-starch fraction (QPS1) [46]. Recent research has also highlighted non-starch polysaccharides (CQNP) extracted using hot water and α-amylase treatment, containing 22.7% carbohydrates, 41.4% protein, and 8.7% uronic acid. Deproteination, enhanced carbohydrate content (39.5%) and altered monosaccharide ratios, shifting from arabinose dominance to galactose predominance, alongside rhamnose and glucose [47].

As illustrated in Figure 2, these polysaccharides from quinoa display notable structural diversity and biological relevance, including the unique monosaccharide profiles and antioxidant potential associated with QPS1 and CQNP.

Additionally, *C. quinoa* Willd microgreens have been identified as a source of complex pectic polysaccharides, specifically homogalacturonan (HG) and rhamnogalacturonan I (RG I). These pectic domains exhibit molecular weights ranging from 2.405 × 10^4^ to 5.538 × 10^4^ Da, illustrating the structural complexity of quinoa’s dietary fibers [48]. In line with these findings, quinoa polysaccharides, such as water-extractable (QWP) and alkali-extractable polysaccharides (QAP), have been shown to contain rhamnose, arabinose, galactose, and galacturonic acid [49].

Beyond seeds, recent research in the variety “Real Blanca” from Bolivia has identified quinoa stalks as a novel source of glucuronoarabinoxylan, a hemicellulose composed of β(1→4)-linked xylose residues substituted with 4-O-methylglucuronic acids, arabinose, and galactose. This high-molecular-weight polymer (700 kDa), offers potential for xylooligosaccharide production using glycoside hydrolases, providing an innovative approach to utilizing quinoa byproducts [50]. Furthermore, Bolivian quinoa’s arabinan polysaccharides, composed of branched (1→5)-linked L-arabinofuranose units, are cross-linked by diferulic acids, which influence their physical properties, such as viscosity and water interaction [51].

In summary, quinoa polysaccharides exhibit diverse structural characteristics and functional properties. Their complexity, ranging from seed-derived arabinan and pectic polysaccharides to stalk-derived glucuronoarabinoxylan, underscores the untapped potential of quinoa as a valuable resource for polysaccharide extraction and utilization. As highlighted in recent research, further investigation into their structural-functional relationships is essential for optimizing their applications in food and health sciences [52]. Figure 2 visually represents the diverse sources and potential uses of quinoa-derived polysaccharides.

### 4.3. Peptides

Quinoa has emerged as a valuable source of bioactive peptides generated through enzymatic hydrolysis of its proteins. These peptides, composed of short amino acid sequences, exhibit diverse structural and physicochemical properties that determine their functionality in various biotechnological and food-related applications.

The composition of quinoa-derived peptides from the altiplano region of Andes is characterized by a balanced profile of essential amino acids, contributing to their stability and bioactivity [53]. The interplay between hydrophilic and hydrophobic amino acids influences their solubility and interaction with biological membranes, affecting their absorption and utilization [54]. Furthermore, peptide length plays a crucial role, with shorter peptides generally exhibiting greater bioactivity due to enhanced absorption and reactivity. Advanced analytical techniques have been employed to determine their molecular weight and sequence, providing insights into their structural properties and chemical interactions. Additionally, purification methods, including ion exchange chromatography, dialysis, and ammonium sulfate precipitation, have facilitated the isolation of bioactive peptide fractions with specific functional properties [55].

A key feature of *C. quinoa* Willd-derived peptides is their ability to self-assemble and form hydrogels, a property with significant implications for food technology and biomedical applications. The peptide KIVLDSDDPLFGGF, identified through the TANGO algorithm, has demonstrated a high propensity for β-aggregation, enabling hydrogel formation under specific conditions. Its structural flexibility allows for conformational transitions between random coil, α-helix, and β-sheet formations, depending on environmental factors such as pH and temperature [56]. High peptide concentrations and pH values near the isoelectric point further promote hydrogel formation, leading to well-defined mechanical properties suitable for functional biomaterials.

In addition to their structural versatility, *C. quinoa* Willd-derived peptides exhibit enzyme inhibitory activity. Using enzymatic hydrolysis using chymotrypsin, protease, and bromelain, bioactive peptides have been identified. Notably, peptides such as QHPHGLGALCAAPPST, HVQGHPALPGVPAHW, and ANDNPSGTVM demonstrated a strong binding affinity to different enzymes related to metabolic regulation [57]. Molecular docking studies confirmed that these peptide–enzyme interactions are mediated by specific amino acid residues, whose charge and hydrophobicity influence binding stability and inhibitory potential.

The production of quinoa protein hydrolysates (QPHs) has been optimized through enzymatic hydrolysis, generating low-molecular-weight peptides with enhanced absorption and biological activity. Characterization using SDS-PAGE and LC-MS has confirmed the generation of bioactive peptides from a protein-enriched fraction (QPF) of *C. quinoa* Willd seeds [58]. Additionally, the hydrolysis process significantly increases amino acid levels, improving the nutritional and functional properties of the resulting peptides. The presence of post-translational modifications, such as phosphorylation and glycosylation, may further influence peptide stability and functionality, although further research is needed to elucidate these aspects [54].

In conclusion, quinoa-derived peptides exhibit a complex chemical structure and a broad range of physicochemical properties that define their functionality in various biotechnological applications. Their enzyme inhibitory activity, self-assembly into hydrogels, and interactions with biological systems highlight their versatility as functional ingredients in food and biomaterials. Figure 2 illustrates the multifaceted bioactivities and structural interactions of quinoa-derived peptides, emphasizing their potential roles in health and technology. Although further in vivo studies are necessary to fully understand their mechanisms and therapeutic potential, current research positions quinoa as a promising source of multifunctional peptides with diverse applications.

## 5. Health Applications of Quinoa Bioactive Compounds

### 5.1. Antimicrobial Activity

Quinoa (*C. quinoa* Willd.) has demonstrated significant antimicrobial activity, largely due to its rich composition of bioactive compounds, such as peptides, saponins, and phenolic acids. Quinoa protein hydrolysates, when enzymatically treated, produce peptides that can inhibit both Gram-positive and Gram-negative bacteria. For instance, studies have shown that these hydrolysates are effective against *Streptococcus pyogenes* and *Escherichia coli*, creating inhibition zones similar to those achieved by gentamicin, a common antibiotic. Under optimal conditions—60 AU/kg protein, a concentration of 800 μg/mL, and incubation at 50 °C for 150 min—the quinoa hydrolysates exhibited inhibition zones up to 12.49 mm [59].

Saponins from quinoa husks have shown potent antibacterial effects, particularly against foodborne pathogens such as *Staphylococcus aureus*, by targeting and disrupting bacterial cell membranes, leading to cell lysis. This membrane-disrupting property makes quinoa saponins promising candidates for natural food preservatives [60]. Phenolic acids and flavonoids in quinoa further add to its antimicrobial potential. Their polyphenolic nature enables them to interact with bacterial cell walls, leading to increased permeability and disruption of cellular processes [61,62].

Interestingly, the antimicrobial efficacy of quinoa appears to vary based on the geographical origin of the crop. Quinoa cultivated in Hongcheon, Korea, and quinoa imported from Peru and the USA have shown distinct bioactive properties. Extracts from seeds of Korean-grown quinoa tend to exhibit strong antioxidant activity, but relatively limited antimicrobial effects, compared to those from imported varieties [63]. This variability suggests that environmental factors, such as soil composition and climate, can impact the development of quinoa’s antimicrobial compounds. Figure 2 highlights how compounds bioactivities, including antimicrobial effects, are central to quinoa’s functional role in food systems.

### 5.2. Antioxidant Activity

Quinoa is highly regarded for its antioxidant properties, attributed to its bioactive compounds, including phenolic acids, flavonoids, saponins, and specific peptides. These antioxidants play a vital role in neutralizing free radicals, thereby reducing oxidative stress, a significant contributor to chronic conditions like heart disease, diabetes, and cancer. Key phenolic compounds such as quercetin, rutin, and vanillic acid are abundant in quinoa and contribute to its ability to protect cellular structures from oxidative damage. Research shows that these compounds can effectively scavenge free radicals, such as DPPH and ABTS, enhancing cellular defense against oxidative stress [64].

A recent research conducted by Zhang and coworkers compared white and black quinoa seeds cultivated in Xining, China, at an altitude of 2300 m, which is comparable to the native Andean range of 2500–4000 m [39]. The results revealed that black quinoa seeds exhibited the highest total phenolic content (up to 643.68 mg/100 g DW), along with the greatest reducing capacity, as determined by ABTS (11.36 µmol TE/g DW) and FRAP (17.14 µmol TE/g DW) assays. This enhanced antioxidant profile was associated with elevated levels of phenolic compounds such as epicatechin, naringenin, and proanthocyanidin B3. In contrast, white quinoa seeds showed the highest total flavonoid content (90.95 mg/100 g DW) and the strongest hydrogen-donating ability, as evidenced by their DPPH activity (91.49 µmol TE/g DW). These properties are likely linked to the presence of hydroxycinnamic acids, including cinnamic acid and its hydroxylated derivatives (2-hydroxy-, 3-hydroxy-, and α-hydroxy-cinnamic acids) [39].

Another study compared the antioxidant capacity across various quinoa plant tissues, including seeds, flowers, leaves, stems, and roots. Their findings indicated that lyophilized flowers presented the highest total phenolic (43 mg GAE/g DM) and flavonoid (12 mg QE/g DM) concentrations, whereas lyophilized seeds exhibited the greatest reducing power according to FRAP analysis (211 mg TE/g DM) compared to air-dried seeds. Additionally, specific antioxidant compounds were distinctly identified in each tissue type, including scopoletin in flowers, rhamnetin 3-glucoside and kaempferol 3-O-sophoroside in seeds, and caffeic acid derivatives in leaves [65].

Additionally, enzymatically hydrolyzed quinoa proteins yield bioactive peptides with enhanced antioxidant properties. Studies on red quinoa protein hydrolysates have shown promising results, demonstrating high radical-scavenging activity both in vitro and in vivo [66]. The antioxidant potential of quinoa polysaccharides is also notable; specific polysaccharides, such as SQAP-2, exhibit excellent radical-scavenging activity comparable to Vitamin C [67]. These polysaccharides are highly effective against various free radicals, including DPPH and hydroxyl radicals, which further supports quinoa’s role as a functional food with antioxidant properties.

The antioxidant activity of quinoa can be enhanced using specific processing methods. For example, microwave treatment of quinoa protein has been shown to improve its digestion rate and antioxidant activity, making it more effective than traditional boiling methods. Supercritical CO_2_ extraction with ethanol has also been used to increase the antioxidant capacity of quinoa protein hydrolysates, highlighting quinoa’s versatility in different forms and potential applications in nutraceuticals [68].

### 5.3. Anticancer and Anti-Inflammatory Activity

Quinoa contains bioactive compounds that exhibit notable anticancer and anti-inflammatory effects. Specific peptides derived from quinoa seeds, such as FHPFPR and NWFPLPR, have demonstrated strong anticancer properties by inhibiting histone deacetylase 1 (HDAC1). This inhibition leads to decreased expression of cancer-promoting genes, including NFκB and Bcl-2, thereby promoting apoptosis in colon cancer cells. These peptides were identified in quinoa seeds of the Mengli-I variety, cultivated in Ulanqab, Inner Mongolia, China, a region characterized by a semi-arid climate and high-altitude conditions that may influence the bioactive profile of the crop. This targeted action on gene regulation makes quinoa peptides potential candidates for anticancer therapies [69].

Polysaccharides extracted from quinoa seeds collected in Jintang (Sichuan, China) have been shown to contribute to its anticancer profile, with selective cytotoxic effects observed against liver (SMMC 7721) and breast (MCF-7) cancer cells. Unlike conventional cancer treatments that may harm healthy cells, quinoa polysaccharides exhibit selective toxicity, sparing normal cells and suggesting a safer alternative for cancer management [70]. The presence of phenolic compounds, including flavonoids like kaempferol and quercetin, further supports quinoa’s anticancer potential. These compounds are known for their ability to inhibit tumor cell proliferation, reduce angiogenesis, and protect cells from DNA damage, making quinoa a valuable component in cancer prevention strategies [61], as shown in Table 1.

In addition to anticancer effects, quinoa exhibits significant anti-inflammatory properties, particularly through its proteins and peptides. The primary protein, chenopodin, extracted from four samples of quinoa seeds (var. Titicaca), has shown efficacy in reducing inflammatory responses in Caco-2 cells by lowering IL-8 expression and inhibiting NF-κB activation when stimulated with inflammatory agents [81]. Protein hydrolysates from white quinoa cultivated in Golmud, Qinghai, China, have been demonstrated to alleviate colitis symptoms in animal models by modulating the gut microbiota and suppressing inflammatory pathways, such as the TLR4/IκB-α/NF-κB signaling pathway [82]. Quinoa leaf extracts, rich in phenolics, have also been shown to reduce nitric oxide production in inflammatory cells, further demonstrating quinoa’s role as a natural anti-inflammatory agent [83].

### 5.4. Additional Health Benefits: Metabolic, Cardiovascular, Immune, and Digestive Health

Beyond its antimicrobial, antioxidant, anticancer, and anti-inflammatory properties, quinoa provides a wide range of health benefits, particularly in metabolic, cardiovascular, immune, and digestive health. Quinoa’s low glycemic index and the presence of specific bioactive peptides contribute to its antidiabetic effects, making it beneficial for individuals with diabetes. These peptides have shown α-glucosidase inhibitory activity, slowing carbohydrate absorption and assisting in blood sugar regulation. Additionally, quinoa-derived peptides have demonstrated antihypertensive effects by inhibiting the angiotensin-converting enzyme (ACE), which plays a key role in blood pressure regulation. This antihypertensive effect supports cardiovascular health and reduces the risk of heart disease [54].

Quinoa’s high fiber and phytosterol content further enhance cardiovascular benefits by aiding in cholesterol management. Phytosterols, plant-based compounds found in quinoa, are known for their ability to reduce low-density lipoprotein (LDL) cholesterol levels, which is essential for maintaining heart health. The healthy fats present in quinoa, such as omega-3 and omega-6 fatty acids, contribute to an optimal lipid profile, supporting vascular health and reducing inflammation in the cardiovascular system [84,85].

Quinoa also offers digestive health benefits, particularly when fermented. Fermented quinoa has an improved nutrient profile, with enhanced mineral bioavailability and reduced anti-nutritional factors like saponins and phytic acid. These modifications make nutrients more accessible, supporting better absorption and digestive function. The fermentation process also supports a balanced gut microbiome by introducing beneficial bacteria, which contribute to gut health by producing short-chain fatty acids (SCFAs). SCFAs help maintain gut integrity and reduce inflammation, providing a foundation for quinoa’s role in promoting digestive health [86], as summarized in Table 1.

In addition to these specific health applications, quinoa’s nutritional density makes it a superfood, providing high-quality proteins, essential amino acids, vitamins, and minerals that are beneficial for overall health. This nutrient density not only supports metabolic health, but also adds to quinoa’s versatility as a dietary staple that can be used to support various therapeutic goals, such as weight management, muscle recovery, and nutrient replenishment [87].

As shown in Table 1, the quinoa data summarize a wide range of bioactive effects including immunomodulatory, anti-inflammatory, hepatoprotective, anti-obesity, hypolipidemic, gut-microbiota-modulating, anticancer, and skin anti-aging activities. Various extracts and fractions from quinoa have been demonstrated to regulate immune responses and metabolic pathways, as evidenced by both in vitro and in vivo studies. 

## 6. Bioactive Compounds in Amaranthus

### 6.1. Phenolic Compounds

Amaranth is a versatile plant renowned for its rich content of phenolic compounds, which impart significant antioxidant and health-promoting properties. These bioactive compounds, distributed across the seeds, leaves, and stems of various *Amaranthus* spp., include both phenolic acids and flavonoids, with concentrations that vary depending on species, environmental factors, and extraction methods. A study involving nine commercially available amaranth samples, sourced from Russia, Peru, and India, analyzed their phenolic acid content. Among the identified phenolic compounds caffeic, ferulic, and p-coumaric acids are noted for their ability to neutralize free radicals [10]. Flavonoids, a subset of phenolic compounds, are also highly abundant in amaranth, with rutin, nicotiflorin, quercetin, and isoquercetin [88,89].

Different species of amaranth showcase unique phenolic profiles. For instance, the ethanolic extract of *Amaranthus retroflexus* leaves contains substantial polyphenolic (0.228 mg/mL) and flavonoid content (2.1 × 10^−4^ mg/mL), surpassing those found in the seeds of other varieties such as “Lera” and “Ultra” [90]. In *Amaranthus tricolor*, the presence of flavonoids like kaempferol and quercetin and phenolic acids like gallic acid contribute to robust biological activities [91]. Similarly, *Amaranthus spinosus* leaves contain an array of phenolic compounds, including flavonoids and tannins, as confirmed using FT-IR analysis [92,93]. In *Amaranthus cruentus*, compounds such as caffeoylsaccharic acid isomer, quercetin 3-O-rhamnosyl-rhamnosyl-glucoside, and kaempferol rutinoside are concentrated in the leaves, with rutin being particularly abundant in the vegetative parts [94]. Figure 3 focuses on the comparative distribution of these phenolic compounds across different *Amaranthus* spp. and plant tissues, highlighting the diversity and relative abundance of flavonoids and phenolic acids. These variations in phenolic compound distribution and concentration across species and plant parts are visually summarized in Figure 3. Germinated amaranth protein concentrate from *Amaranthus hypochondriacus* further enhances bioactivity by releasing phenolics like methyl gallate and syringic acid [95].

A recent study analyzed 120 accessions across the leaves of 9 *Amaranthus* species and found substantial diversity in the content of 17 polyphenolic compounds. Rutin emerged as the dominant compound, with levels varying significantly between species and years. Strong positive correlations were observed among certain compounds (e.g., rutin and kaempferol-3-O-β-rutinoside), while gallic acid showed consistent negative correlations. Variations across years and accessions highlight the influence of both genetic and environmental factors [96].

### 6.2. Peptides

Amaranth peptides are derived from the storage proteins found in amaranth grains. These proteins contain encrypted amino acid sequences that, once released through enzymatic hydrolysis, exhibit a range of physiological activities [97]. Amaranth is particularly valued for its high-quality protein content and complete amino acid profile, including lysine, which is typically limited in other cereals [98]. The main storage proteins are globulins, with globulin 11S—also known as amarantin—being a principal source of bioactive peptides. During food processing or digestion, proteolytic cleavage of these proteins liberates peptides with potential health-promoting effects.

Several studies have identified bioactive peptides from Amaranthus protein hydrolysates, with molecular weights ranging from 500 Da to 1.44 kDa. These peptides are relatively small, favoring intestinal absorption and enhancing bioactivity. For example, the peptide ITASANEPDENKS (1.44 kDa) was among the largest identified, while others ranged between 8 and 14 amino acids in length [99]. Their sequences frequently contain hydrophobic amino acids such as leucine, valine, proline, and isoleucine, which may facilitate interaction with membrane-bound enzymes such as ACE or HMG-CoA reductase, suggesting antihypertensive and hypocholesterolemic potential [98,100].

Using sequence data from databases like UniProt and predictive tools such as BIOPEP, researchers have mapped fragments from amaranth proteins with predicted ACE-inhibitory, DPP-IV inhibitory, and antioxidant properties [100]. These peptides often include residues like tryptophan and tyrosine, known for stabilizing free radicals and enhancing antioxidant effects. Acidic amino acids (e.g., aspartic and glutamic acid) contribute to metal chelation, further supporting their antioxidant role.

Among the peptides identified through in vitro hydrolysis of *Amaranthus cruentus*, GGV, IVG/LVG, and VGVI/VGVL were notable for their small size (all < 3 kDa) and high hydrophobic content. These features may promote interaction with lipid membranes and key enzymes involved in metabolic regulation [98].

A detailed study on the gastrointestinal digestion of 11S globulin revealed ten bioactive peptides, mainly from the acidic subunit, with sequences such as TEVWDSNEQ, IYIEQGNGITGM, and KFNRPETT. These peptides showed molecular weights around 1 kDa. GRAVY index analysis indicated that most were hydrophilic, although some exhibited amphipathic characteristics. Their net charge at physiological pH, combined with features like terminal arginine residues, may modulate their interaction with cell membranes or molecular targets [100]).

Together, these findings highlight the potential of amaranth-derived peptides as multifunctional bioactive agents for use in nutraceutical and functional food applications.

### 6.3. Polysaccharides

*Amaranthus* spp. has been extensively studied for its polysaccharide content, highlighting its structural diversity, functional properties, and health-promoting potential. Pectic polysaccharides, as the primary polysaccharides in *A. caudatus*, play a crucial role in plant cell structure and exhibit biological activities. These polysaccharides consist of galacturonic acid, rhamnose, and galactose, with complex structural features, including specific molecular weights and functional groups, which underscore their significance in food and health applications [101].

As described in *A. hybridus* L., polysaccharides extracted using microwave-assisted methods were purified into two fractions, AHP-M-1 and AHP-M-2, with molecular weights of 77.625 kDa and 93.325 kDa, respectively [102]. Similarly, *A. hybridus* polysaccharides (AHP-H-1 and AHP-H-2) exhibited distinct monosaccharide compositions, analyzed using GC-MS, which revealed structural differences likely to influence their biological activity and functional properties, emphasizing the importance of understanding this diversity [103].

*A. retroflexus* contains a high polysaccharide content (39.4% of its dry mass), including 2% water-soluble polysaccharides that play vital roles in physiological processes. Additionally, it features notable levels of pectin, essential for plant structure and digestive health, and hemicellulose B (23.3%), contributing to cell wall integrity and nutritional value. The primary monosaccharides identified in its water-soluble polysaccharides include arabinose, glucose, and galacturonic acid, which are critical for their functional properties [104].

The pectic polysaccharides in *A. cruentus* have also been extensively studied. These include a linear α-1,4-D-galacturonan backbone and branched rhamnogalacturonan regions, which enhance their gelling ability and molecular interactions. Neutral sugars such as rhamnose, arabinose, galactose, and xylose contribute to their structural complexity and functional potential. Novel methods, such as using rotary impulse apparatus, have facilitated the extraction of these polysaccharides, which form water-soluble complexes with macro- and microelements [105].

*A. tricolor* L. has demonstrated additional promise as a source of polysaccharides. Heteropolysaccharides extracted from its stems were composed of glucose, galactose, and rhamnose, with structural analysis via FTIR and NMR revealing functional groups and glycosidic linkages. Molecular weight determination further highlighted their potential for applications in food, pharmaceuticals, and other industries, especially in influencing viscosity and gelling properties [106]. Moreover, a study on its leaves identified 48 monosaccharides and 28 derivatives, with young leaves showing higher hydrophilic carbohydrate metabolite concentrations. These metabolites suggest adaptive potential, but require further research to explore polysaccharide-specific properties [107].

A broader review emphasizes the importance of soluble dietary fibers, resistant starch, pectins, and hemicelluloses in *Amaranthus* spp. The unique chemical structures of amaranth polysaccharides are central to their functional properties and diverse applications in food science and nutrition [108].

In conclusion, the extensive structural diversity and functional potential of amaranth polysaccharides position them as valuable resources for applications in health, nutrition, and industry. Continued research into their composition and biological activities will further elucidate their roles and optimize their applications across various fields.

## 7. Health Applications of Amaranthus

### 7.1. Antimicrobial Activity

Species within the *Amaranthus* genus exhibit significant antimicrobial activity, attributed to their rich phytochemical composition, including flavonoids, phenolic acids, and other bioactive compounds. Various studies have evaluated their potential to inhibit bacterial and fungal pathogens, demonstrating promising results that support their potential as natural antimicrobial agents.

A recent study highlighted the strong antimicrobial activity of methanolic extracts from *A. spinosus* leaves, collected from uncultivated farmland in Awo-Omamma, Nigeria. The minimum bactericidal concentration (MBC) was 1.00 mg/mL against *Bacillus* spp., *E. coli*, and *Pseudomonas aeruginosa*, while the minimum fungicidal concentration (MFC) was 0.125 mg/mL against *Candida albicans* and *Aspergillus niger*. These results, comparable to standard antibiotics such as ciprofloxacin and the antifungal ketoconazole, suggest that *A. spinosus* extracts could serve as effective antimicrobial agents [109].

Similarly, methanolic extracts from the aerial parts of *Amaranthus viridis*, collected from six different districts of the Fayoum Depression in Egypt exhibited significant antimicrobial activity in 96-well plate assays. Specific fractions demonstrated notable effects against *S. aureus*, *Salmonella typhi*, and *E. coli*, with strong biofilm inhibition activity observed for AV4 and AV5 against *E. coli* and *Bacillus subtilis*, while AV6 was particularly effective against *P. aeruginosa*. The antifungal assays indicated that AV6 was highly active against *Aspergillus flavus* and *A. niger*, whereas AV2 and AV5 displayed minimal inhibition of *C. albicans*. These findings highlight *A. viridis*’ potential in treating biofilm-associated infections [110]. These antimicrobial effects, alongside species-specific differences in bactericidal and fungicidal activity, are summarized in Figure 3, which highlights the main bioactivities of various *Amaranthus* spp.

Another study examined the antifungal activity of mature leaves from various *Amaranthus* spp. against *Fusarium equiseti*, *Rhizoctonia solani*, *and Alternaria alternata.* The species tested included *Amaranthus* spp., *A. cruentus*, and *A. hypochondriacus × hybridus* cultivated in Bodaczów near Zamość (southeastern Poland), as well as *A. retroflexus* and *A. hybridus* from plantations in the Düzce province (northwestern Turkey). *A. hybridus* exhibited the strongest effect due to its high polyphenol content. However, no inhibitory effect was observed against *Fusarium oxysporum* and *Colletotrichum coccodes*. The extracts primarily displayed fungistatic activity, particularly in the initial days of exposure, suggesting a selective and short-term antifungal effect [111].

Additionally, ethanolic extracts from the whole plant of *A. tricolor*, cultivated in the botanical garden of Hezhou University in China, exhibited notable antimicrobial properties, particularly in the ethyl acetate (EtOAc) fraction. Three key bioactive compounds were identified, with gallic acid exhibiting the highest antimicrobial activity against bacteria and fungi, with MIC values ranging from 63 to 68 μg/mL. In particular, *A. tricolor* extracts were highly effective against *E. coli* and *S. aureus*, suggesting their potential for applications in food preservation and health [91].

The antimicrobial effects of *A. caudatus* were also evaluated using ethanol and aqueous extracts at different growth stages (including shoots, flowering, and pre-flowering stages). The extracts exhibited significant inhibition against *C. albicans*, suggesting potential antifungal applications. Mild inhibitory effects were also observed against *Candida glabrata*, *Penicillium chrysogenum*, and *Penicillium aurantiogriseum*. Interestingly, the antimicrobial activity varied based on soil composition, with ethanolic extracts from plants grown in clayey loam soil exhibiting higher toxicity, particularly at the post-flowering stage [112].

Crude extract from *A. tricolor*, prepared from intact leaves obtained at a local market in Heilongjiang province, China, demonstrated potent antimicrobial activity against *S. aureus*, with inhibition zone diameters ranging from 12.63 ± 0.34 mm to 12.94 ± 0.43 mm and an MIC of 80 mg/mL. The study identified multiple mechanisms of action, including membrane depolarization, intracellular pH reduction, decreased bacterial protein content, DNA cleavage, and cytoplasmic leakage, confirming its ability to effectively inhibit bacterial growth [113].

Finally, a comparative study of *A. spinosus* and *Tridax procumbens* revealed significant antimicrobial properties against various bacterial strains, including antibiotic-resistant ones. Hydroethanolic extracts from *T. procumbens* exhibited the highest activity, with inhibition zones ranging from 9 to 31 mm, except against *C. albicans* 1581. Meanwhile, *A. spinosus* inhibited 11 of the 15 tested strains, with inhibition zones between 7 and 17.5 mm. MIC values ranged from 0.39 to 3.12 mg/mL, with *T. procumbens* showing superior potency, underscoring its potential for therapeutic applications [114].

These studies collectively reinforce the potential of *Amaranthus* spp. as a natural source of antimicrobial agents with applications in both medicine and food preservation. Antimicrobial activity varies depending on the species, extract type, and growth conditions, suggesting that future research should focus on optimizing their use by identifying and characterizing the most relevant active compounds.

### 7.2. Antioxidant Activity

The antioxidant potential of *Amaranthus* spp. is attributed to their diverse phytochemical composition, which includes vitamins, tocopherols, polyphenols, and squalene, compounds known for their ability to counteract oxidative stress [115,116]. Several studies have explored the antioxidant properties of amaranth, analyzing its bioactive compounds, processing effects, and mechanisms of action.

Cooking methods significantly influence the antioxidant activity of amaranth grains. A comparative study between boiling and roasting revealed that roasting enhances the total phenolic content (TPC) and antioxidant properties, whereas boiling leads to higher starch hydrolysis, but reduces antioxidant potential. Specifically, roasted intact grains exhibited a TPC of 662.78 mg GAE/100 g sample, compared to a 470.21 mg GAE/100 g sample in boiled grains, underscoring the potential of roasting to improve amaranth’s antioxidant capacity [89].

The antioxidant potential of four *Amaranthus* spp.—*A. viridis*, *A. spinosus*, *A. tricolor*, and *A. lividus*—was evaluated using DPPH and ABTS assays. Among them, *A. viridis* demonstrated the highest radical scavenging activity, suggesting its superior ability to mitigate oxidative stress. *A. spinosus*, while recognized for its high protein and fiber content, exhibited moderate antioxidant activity. *A. tricolor* and *A. lividus* contributed bioactive pigments and nutritional value, but did not display antioxidant potential as high as *A. viridis*. The study highlighted *A. viridis* and *A. tricolor* as underutilized yet promising sources of antioxidants for health benefits [89]. These interspecific variations in antioxidant capacity, related to phenolic content and processing methods, are also illustrated in Figure 3, offering a visual synthesis of key functional traits.

Further research assessed the antioxidant properties of amaranth flour and beverage following simulated gastrointestinal digestion. Peptides generated from digested amaranth flour and beverage exhibited notable peroxyl scavenging activity (ORAC). The beverage-derived bioaccessible fractions demonstrated superior ORAC potency compared to flour-derived bioaccessible fractions, suggesting that processing methods influence peptide composition and antioxidant efficacy [117].

An extensive evaluation of 16 *Amaranthus* accessions investigated their morphological and biochemical traits, including antioxidant content, to determine their suitability for cultivation in Central Russia. The study revealed significant variability in water-soluble antioxidant content, ranging from 0.425 to 1.439 mg GAE/g fresh weight (FW). The Valentina cultivar, characterized by red leaves, exhibited the highest concentration of amaranthine (0.319–2.031 mg/g FW), a potent antioxidant compound. Additionally, total phenolic content in leaves varied between 2.700 and 4.825 g GAE/g FW, highlighting the strong antioxidant potential across different cultivars [118].

Leaves, roots, and stems of *A. spinosus* L., collected from local harvests in the Hooghly district of the Gangetic plain, West Bengal, India, were subjected to phytochemical analysis. This investigation identified 21 bioactive polyphenolic antioxidants derived from the phenylpropanoid pathway, which contribute to oxidative stress reduction and disease prevention. In vitro assays confirmed the plant’s significant antioxidant activity, including free radical scavenging, metal-chelating, reducing power, and scavenging superoxide and hydroxyl radicals [93].

In a biological model study, amaranth leaves sourced from a local vendor in Uganda were analyzed, and the *Amaranthus* leaf extract demonstrated protective effects against hydrogen peroxide-induced oxidative stress in *Drosophila melanogaster*. Flies consuming 0.1 mg/mL of *Amaranthus* extract exhibited significantly higher survival rates under oxidative stress conditions, suggesting a dose-dependent protective effect. The antioxidant activity of *Amaranthus* leaves was attributed to their high ascorbic acid and polyphenol content, which effectively neutralize free radicals [119].

The nutritional and antioxidant potential of *A. tricolor*, *A. viridis*, and *Achyranthes aspera* was further explored using GC-MS profiling of volatile metabolites. The extracts exhibited substantial antioxidant activity, with EC50 values for free radical scavenging ranging from 34.1 ± 1.5 to 166.3 ± 14.2 µg/mL and ferric reducing antioxidant power (FRAP) values between 12.1 ± 1.0 to 34.0 ± 2.0 µg Trolox Equivalent/mg. These findings underscore their potential role in oxidative stress management and dietary health applications [120].

Additionally, enzymatic hydrolysis of amaranth stubble using Flavourzyme^®^ and Alcalase^®^ revealed the presence of bioactive peptides with antioxidant properties. The degree of hydrolysis achieved was 16.31% with Flavourzyme^®^ and 12.64% with Alcalase^®^ resulting in peptides ranging from <1 kDa to >10 kDa. These hydrolysates exhibited strong antioxidant capacity, suggesting their potential application as natural antioxidants in food preservation and nutraceutical formulations [115]. 

Collectively, these studies confirm the significant antioxidant potential of *Amaranthus* spp., with variations depending on species, processing methods, and extraction techniques. These findings support the incorporation of amaranth into functional foods and health products to mitigate oxidative stress and promote overall well-being.

### 7.3. Anticancer and Anti-Inflammatory Activity

Studies on the *Amaranthus* genus have demonstrated its anticancer and anti-inflammatory potential due to its rich composition of bioactive compounds, as detailed in Table 2. *A. spinosus Linn.* exhibits anticancer activity against breast, hepatocellular, prostate, and colorectal cancers, attributed to its glycosides, phenolic compounds, and terpenoids. Additionally, it possesses anti-inflammatory and antioxidant properties that enhance its therapeutic applications [121]. Similarly, *A. viridis* has been identified as a source of flavonoids, triterpenes, and phenols with anticancer and anti-inflammatory effects, with key compounds such as quercetin, kaempferol, and vitexin contributing to its biological activity [122]. These findings support the traditional use of these species in herbal medicine and highlight the need for further studies to elucidate their mechanisms of action [123].

Beyond its anticancer potential, *Amaranthus* has been recognized for its ability to modulate inflammatory processes, a key factor in cancer progression. Studies have shown that its phytochemicals, including quercetin, rutin, apigenin, and squalene, can inhibit the biosynthesis of COX and LOX enzymes, reducing the production of pro-inflammatory eicosanoids such as prostaglandins and leukotrienes [134]. This anti-inflammatory activity has been confirmed in animal models, such as the ethanolic root extract of *A. caudatus* from the area of Moradabad, India, which exhibited inflammation inhibition comparable to diclofenac sodium [135]. Additionally, a study on kiwicha (*Amaranthus caudatus* L.) protein hydrolysates demonstrated significant anti-inflammatory effects in vitro. Using the Caco-2 cell line, researchers found that these hydrolysates decreased the expression of pro-inflammatory cytokines, increased anti-inflammatory cytokines, and reduced the gene expression of key inflammasome components [125]. These findings suggest that *A. caudatus* hydrolysates may serve as a valuable source of bioactive peptides with potential applications in functional foods and nutraceuticals aimed at reducing intestinal inflammation. Similarly, *A. spinosus* and *A. viridis* have shown significant effects in regulating inflammatory mediators, reinforcing their therapeutic potential for chronic inflammatory diseases [136,137]. As shown in Figure 3, these effects are associated with specific bioactive compounds such as quercetin, vitexin, and squalene, which contribute to both anti-inflammatory and anticancer mechanisms.

The role of *Amaranthus* in cancer prevention has been supported by specific model studies. In a DMBA-induced breast cancer model, the leaves of *A. hybridus* and Corchorus olitorius exhibited chemopreventive effects by modulating estradiol levels and reducing inflammation through IL-6 suppression, with a synergistic interaction between both species [138]. Meanwhile, *A. hypochondriacus* demonstrated cytotoxic activity against small-cell lung cancer (H69V) and hepatocellular carcinoma (HepG2/C3A), with an inhibitory concentration of 70.55 µg/mL, positioning it as a promising source of anticancer agents [133].

The therapeutic potential of *Amaranthus* extends beyond its effects on chemotherapy-sensitive tumor cells to resistant cancer models. A study on AS20, a herbal formulation derived from *A. spinosus*, demonstrated its ability to induce apoptosis in 5-Fluorouracil-resistant HeLa cells by downregulating COX-2 and modulating the expression of apoptotic genes such as BAX, BCL2, and BCL2L1 [139]. These findings suggest that *Amaranthus* extracts could serve as adjuncts in cancer treatment, enhancing chemotherapy efficacy and reducing inflammation associated with tumor progression.

In conclusion, the *Amaranthus* genus emerges as a promising source of therapeutic agents with anticancer and anti-inflammatory properties. Its activity against various cancers, ability to inhibit key inflammatory pathways, and relative safety reinforce its value in natural medicine and pharmaceutical research. However, further studies are necessary to isolate and characterize the compounds responsible for these effects, as well as to evaluate their safety and efficacy in clinical models.

### 7.4. Additional Health Benefits: Metabolic, Cardiovascular, Immune, and Digestive Health

Squalene and tocopherols have demonstrated cholesterol-lowering properties by inhibiting cholesterol absorption and enhancing antioxidant activity. Studies have shown that regular consumption of amaranth oil can significantly reduce LDL levels and improve overall lipid profiles [140]. Additionally, polyphenols like rutin and caffeic acid, contribute to the anti-inflammatory and immunomodulatory effects of amaranth by blocking NF-κB signaling and reducing pro-inflammatory cytokine production [141]. Lunasin also activates tumor suppressor proteins and mitigating oxidative stress [142].

Peptides derived from amaranth proteins have been shown to inhibit dipeptidyl peptidase-IV (DPP-IV), an enzyme involved in glucose metabolism, positioning amaranth as a promising functional food for managing type 2 diabetes [143,144]. The seeds are also a notable source of calcium, magnesium, and phosphorus, which support bone mineral density and may help prevent osteoporosis, particularly in individuals with celiac disease [145]. The hydromethanolic extract of amaranth seeds has also shown the ability to decrease nitric oxide (NO) production in RAW 264.7 macrophage-like cells while increasing nitrate (NO_3_^−^) and nitrite (NO_2_^−^) levels in vivo, eight hours after a single oral dose [146,147]. This modulation of NO metabolism suggests an enhanced performance response, potentially beneficial for individuals engaged in vigorous physical activities or sports. Additionally, selenium and betacyanins in edible amaranth seed sprouts have demonstrated anti-inflammatory effects by preventing NF-κB translocation to the nucleus and significantly reducing pro-inflammatory cytokine IL-6 levels in RAW 264.7 macrophages [148].

Amaranth’s benefits extend to metabolic regulation, as its grain and oil fractions have been shown to significantly decrease serum glucose levels and increase serum insulin levels in diabetic rats, indicating its potential for correcting hyperglycemia and preventing diabetic complications [141]. However, the specific components responsible for these anti-obesity and anti-diabetic effects require further investigation. While most studies on amaranth’s metabolic benefits have been conducted in vitro or in animal models, human clinical trials are essential to validate these findings [141]. In human studies, daily intake of 18 mL of amaranth oil for three weeks was found to significantly lower total cholesterol, triglycerides, LDL, and very-low-density lipoprotein (VLDL) cholesterol levels [149]. Elevated LDL levels above 130 mg/dL, high-density lipoprotein (HDL) cholesterol below 35 mg/dL, and total blood cholesterol levels above 200 mg/dL are considered indicators of heightened cardiovascular risk. Amaranth seeds, similar to quinoa, contain high-quality polyunsaturated fatty acids, lutein, and tocopherols, which may contribute to cardiovascular health benefits, although further studies are needed to confirm these effects in human populations [150,151].

This comprehensive health profile underscores the potential of amaranth as a key ingredient for the development of functional foods and nutraceuticals aimed at preventing chronic diseases and improving overall health [142].

As shown in Table 2, the amaranth findings reveal significant anti-inflammatory, immunomodulatory, antidiabetic, hepatoprotective, and anticancer properties. Extracts and protein hydrolysates from different amaranth sources have been shown to modulate cytokine expression, oxidative stress, and metabolic enzyme activity, with supportive evidence from both in vitro and in vivo models.

## 8. Bioactive Compounds in *Lupinus*

### 8.1. Polyphenols

*Lupinus* spp. are a rich source of polyphenolic compounds, with significant variations in composition, concentration, and bioactivity across different species and plant parts. The application of advanced analytical techniques such as LC-HRMS, UHPLC-ESI-MS/MS, HPLC-DAD-ESI-MS/MS, and ^1^H NMR-based metabolomics has allowed for the precise identification, quantification, and characterization of these bioactive molecules.

The polyphenol content in *L. albus* seeds cultivated in Southern Italy was analyzed using LC-HRMS, identifying 14 different polyphenols with ellagic acid (16,271.86 ± 19,798.53 μg/Kg) as the most abundant, followed by apigenin (2749.51 ± 889.95 μg/Kg). These findings revealed that environmental factors influence polyphenol variability [152]. Similarly, an analysis of *L. albus* using ^1^H NMR and UHPLC-ESI-MS/MS identified flavonoids and phenolic acids. A comparative analysis with other legumes emphasized the unique polyphenolic profile of *L. albus*, further supporting its role as a functional food ingredient [153].

The seed coat of *L. angustifolius* was investigated using HPLC-DAD-ESI-MS/MS, revealing a high polyphenol concentration comparable to or exceeding other common sources of polyphenols. This study reinforces the nutritional and functional relevance of the lupin seed coat in food applications [154]. Additionally, *L. angustifolius* seeds exhibited the highest total polyphenol content (696.21 mg GAE/100 g) among four *Lupinus* spp., followed by *L. albus* (614.13 mg GAE/100 g), *L. luteus* (467.78 mg GAE/100 g), and *L. mutabilis* (367.36 mg GAE/100 g) [155].

In *Lupinus luteus* germinated seeds produced by Plantico (Waganiec, Poland), apigenin and fisetin were identified as the dominant flavonoids. The study also identified 20 different phenolic compounds, including salicylic, ferulic, and sinapinic acids, and highlighted the presence of glucosides, which may influence bioactivity. Polyphenol content increased significantly during germination, with peak concentrations observed after six days, suggesting that germination enhances the bioactive potential of *L. luteus* [156].

Polyphenolic profiling of *L. mutabilis* seeds provided by the Leguminous Program of the Universidad Nacional Agraria La Molina, Lima, Perú, indicated that flavonoids account for 85.5–99.6% of free phenolic compounds, with key molecules including genistein and its derivatives, apigenin, catechin, and naringenin. Smaller quantities of phenylethanoids (e.g., tyrosol derivatives) and cinnamic acid derivatives were also identified. Among different ecotypes, the H6 INIA BP variety exhibited the highest total free phenolic concentration (1393.32 mg/kg DM) [153].

The roots of *Lupinus polyphyllus* were found to contain a diverse range of phenolic compounds, including (−)-epicatechin (213.05 µg/mL), (+)-catechin (208.06 µg/mL), and (−)-epigallocatechin (154.95 µg/mL). The HPLC-based quantification confirmed that 1.972% of root biomass consists of phenolic compounds, indicating the presence of condensed tannins, which contribute to antioxidant activity [157].

Beyond profiling, several studies examined structural stability and bioactivity modulation of *Lupinus* polyphenols. Research on heat-treated *L. albus* seeds showed that thermal processing slightly increased total polyphenol content (from 6142.50 mg GAE/100 g to 6418.44 mg GAE/100 g), demonstrating that heat exposure does not significantly degrade polyphenols. Furthermore, the study highlights the presence of macromolecular polyphenols, which contribute to greater antioxidant capacity [158]. Key phenolic acids distributed across *Lupinus* spp. include p-hydroxybenzoic acid, protocatechuic acid, vanillic acid, syringic acid, and hydroquinone, alongside simpler phenols like catechol, orcinol, phloroglucinol, and pyrogallol. These compounds are essential in understanding flavonoid biosynthesis in *Lupinus* [159].

The polyphenol composition of *Lupinus* spp. varies significantly across species, plant parts, and environmental conditions. The identification and quantification of polyphenols using advanced analytical methods have provided valuable insights into their structural diversity, antioxidant potential, and functional properties. Notably, flavonoids such as apigenin, fisetin, genistein, catechin, and naringenin play a dominant role in bioactivity, while thermal processing and germination influence polyphenol stability and concentration. These findings establish *Lupinus* as a promising source of dietary polyphenols, with applications in functional foods, nutraceuticals, and health-related formulations.

### 8.2. Peptides

*Lupinus*-derived peptides have been extensively studied for their structural properties, isolation techniques, and biochemical functions, revealing their potential in diverse biological applications. The enzymatic hydrolysis of *L. mutabilis* proteins, performed using pancreatin, papain, and endopeptidase, has allowed for the extraction of peptides with distinct molecular weights and bioactivities. These peptides were fractionated into Lupin Protein Hydrolysate (LPH) 1 (>10 kDa), LPH2 (3-10 kDa), and LPH3 (<3 kDa), with LPH3 demonstrating the highest functional activity [160]. The identification and structural characterization of these peptides have been achieved using mass spectrometry (MS) and high-performance liquid chromatography (HPLC), providing valuable insights into their sequence composition and biochemical behavior.

Among the many lupin seed-derived bioactive peptides identified, AVPFWM, YSGWLGL, AHAGFGMLY, and FFSMKVM emerged as the most bioactive [161]. Their small molecular size and hydrophobic amino acid composition contributed to their increased bioavailability and biological activity [162]. These findings reinforce the structural importance of peptide sequences in determining their functional roles, particularly in metabolic and cardiovascular applications.

Beyond *L. mutabilis*, other species of *Lupinus* have also been explored for their peptide composition. α/γ-Conglutins, major seed storage proteins, have been identified as precursors for bioactive peptides with diverse metabolic functions. Using enzymatic hydrolysis, these proteins generate peptides that have been extensively analyzed using Fourier-transform infrared (FTIR) spectroscopy and nuclear magnetic resonance (NMR) spectroscopy, techniques that provide insights into their secondary structures and folding patterns [163]. This structural characterization is crucial, as peptide conformation influences their stability, receptor binding, and overall bioactivity.

One particularly well-characterized peptide, GPETAFLR, is an α-helical octapeptide isolated from *Lupinus angustifolius* L., a leguminous species whose seeds were obtained from Koipesol Semillas, S.A. (Seville, Spain). The peptide sequence—comprising glycine, proline, glutamic acid, threonine, alanine, phenylalanine, leucine, and arginine—confers a stable α-helical structure that is thought to enhance its interaction with specific biological targets. [164]. These findings highlight the role of peptide sequence specificity in determining bioactivity, further reinforcing the importance of detailed structural characterization.

Interestingly, the role of peptides in plant development has also been explored. The identification of LaCEP1, a C-terminally encoded peptide (CEP) family member in *Lupinus albus*, revealed its function in modulating cluster root formation. Further experiments revealed that a 15-mer core peptide derived from LaCEP1 was sufficient to induce morphological changes, confirming its functional role in root architecture regulation [165]. This study extends the significance of Lupinus-derived peptides beyond human health, emphasizing their role in plant physiology and adaptation to nutrient stress.

Taken together, these findings—along with the broad biological activities summarized in Figure 4—illustrate the multifaceted nature of *Lupinus* peptides, spanning metabolic regulation, structural stability, and even plant development. The use of advanced biochemical techniques, such as MS, HPLC, FTIR, and NMR, has been pivotal in elucidating peptide sequences, molecular structures, and their interactions with biological systems. As research continues, further characterization of these peptides may lead to novel applications in functional foods, nutraceuticals, and plant biotechnology.

### 8.3. Polysasccharides

The polysaccharides present in *Lupinus* seeds are structurally diverse and biochemically complex, encompassing both soluble and insoluble dietary fibers. In this study, lupin bean powder *(Lupinus angustifolius;* TLC Lupin Flakes) was supplied by The Lupin Co. (Fremantle, WA, Australia) as the source material. The soluble fraction includes low-molecular-weight oligosaccharides such as stachyose and verbascose, as well as dominant pectic polysaccharides like homogalacturonan and rhamnogalacturonan I (RGI), which are enriched with (arabino-)galactans and branched arabinans. The insoluble fraction primarily consists of cellulose, pectins, xylans, and highly substituted fucosylated xyloglucans, contributing to the structural integrity of the plant cell wall [166].

In *Lupinus angustifolius*, the main soluble polysaccharide extracted—referred to as LuPS-8—is a linear pectic polymer with a high molecular mass of 6608 kg/mol. This polymer was isolated from lupin bean powder supplied by The Lupin Co. (Australia). Its composition is predominantly galactose (71.0%), with lesser amounts of arabinose (16.0%), glucuronic acid (4.6%), and galacturonic acid (4.1%). These structural features support its potential functional roles in food systems and health applications [167].

Lupin kernel fiber is largely composed of non-starch polysaccharides, which are resistant to human digestion and contribute significantly to the dietary fiber content. These include cellulose, non-cellulosic glucans, and pectin-like substances that enhance solubility and potential health benefits. In addition, raffinose family oligosaccharides (raffinose, stachyose, and verbascose) are present, with stachyose being the most abundant, particularly in *L. hispanicus*, where it accounts for up to 75% of the total oligosaccharide content. These compounds may have implications for seed metabolism and biological function [168].

Further structural characterization of lupin seed polysaccharides across multiple accessions has revealed the presence of monosaccharides such as D-galactose, L-arabinose, D-galacturonic acid, D-xylose, and L-rhamnose in varying molar ratios (15:10:6:2:2 and 6:4:3:1:1). These findings, derived from the comparative analysis of 33 Lupinus seed accessions and cultivars representing 10 different species, highlight potential interspecific variability that may influence the functional properties and bioactivity of these polysaccharides [169].

In-depth studies on *L. angustifolius* seeds—sourced from Coorow Seeds (Coorow, WA, Australia)—using hot water extraction and size-exclusion chromatography led to the identification of six polysaccharide fractions (BLP-1 to BLP-6). These fractions exhibited diverse compositions, including galactose, fucose, rhamnose, glucose, mannose, ribose, and xylose. A notable finding was the direct correlation between galactose content and antioxidant activity, with BLP-1, BLP-2, and BLP-5 displaying the highest radical scavenging potential [170].

A comprehensive analysis of *L. angustifolius* and *L. luteus* focused on the extraction and characterization of pectins and hemicelluloses. Using water, oxalic acid, and potassium hydroxide, researchers obtained water-soluble pectins (5.0–5.5 g), acid-soluble pectins (1.5–2.0 g), and uronide-rich hemicelluloses (8.0–9.5 g) from *L. angustifolius* biomass [171].

Taken together, these findings underscore the structural complexity and biochemical richness of lupin-derived polysaccharides. The use of advanced analytical tools—including ^13^C NMR, IR spectroscopy, chromatography, and electrophoresis—has been essential in unraveling their molecular architecture and functional attributes. This growing body of knowledge enhances our understanding of lupin polysaccharides and supports their potential applications in functional foods, biopolymer development, and industrial biotechnology.

## 9. Health Applications of *Lupinus*

### 9.1. Antimicrobial Activity

The antimicrobial potential of *Lupinus* spp. has been widely studied, revealing the presence of various bioactive compounds, including hydrolysates, alkaloids, proteins, peptides, and nanoparticle-synthesizing metabolites. These components exhibit significant antibacterial and antifungal activities, positioning *Lupinus* as a promising natural source of antimicrobial agents for pharmaceutical, food preservation, and biotechnological applications.

The antimicrobial activity of *Lupinus* hydrolysates is strongly influenced by the type of flour and the enzymatic hydrolysis process. Notably, hydrolysates derived from debittered and defatted flour demonstrated enhanced antimicrobial effects compared to those from raw flour. Moreover, optimal hydrolysis conditions—specifically the use of pancreatin for 48 h—maximized the inhibition of *S. aureus* and *E. coli*, suggesting their potential as natural antimicrobial agents in food preservation and health applications [172]. In a similar vein, enzymatic hydrolysis and fermentation have been used to generate bioactive peptides from *Lupinus* seeds, which have shown strong antibacterial activity against both Gram-positive *(S. aureus*, *B. subtilis*) and Gram-negative (*P. aeruginosa*, *E. coli*) bacteria. These peptides exert their antimicrobial effects by disrupting bacterial membranes and interfering with essential cellular processes, with some fractions demonstrating over 85% inhibition against specific bacterial strains [161].

In addition to peptides, *Lupinus* seeds contain a variety of alkaloids with significant antimicrobial potential. A study analyzing 22 genotypes of *Lupinus* spp. (*L. albus*, *L. luteus*, *L. angustifolius*) identified more than 30 alkaloids, with notable variations among species and genotypes. Extracts from these seeds displayed potent antibacterial activity against *K. pneumoniae* and *P. aeruginosa*, suggesting their potential in combating infections caused by these pathogens. Interestingly, *L. albus* landraces exhibited the highest alkaloid concentrations, with major compounds including lupanine, 13α-hydroxylupanine, and albine, whereas *L. angustifolius* primarily contained lupanine, 13α-hydroxylupanine, and angustifoline [173]. The presence of these bioactive alkaloids underscores the pharmaceutical potential of *Lupinus* as a natural antimicrobial source.

Beyond seeds, Lupinus leaf and root extracts have also demonstrated antimicrobial effects. *Lupinus arboreus* leaf extract and its fractions, obtained through cold maceration with methanol, exhibited strong antibacterial activity, particularly against *B. subtilis* and *S. aureus*, although its antifungal effects were weaker. Among the fractions tested, the ethyl acetate fraction showed the highest antimicrobial potency, highlighting its potential as a source of novel bioactive compounds for therapeutic applications [174]. In contrast, while *Lupinus termis* root oil extract lacked antimicrobial activity, its methanolic extract, rich in quinolizidine alkaloids, demonstrated moderate antibacterial effects. These findings, alongside the traditional use of *L. termis* seed extracts in ethnopharmacology for treating shingles, reinforce the medicinal significance of *Lupinus* spp. [175].

Protein fractions from *Lupinus* seeds have also been identified as key contributors to their antimicrobial properties. Specifically, 11S globulins and a basic subunit (BS) from *L. termis* seeds demonstrated significant antibacterial effects against *Listeria monocytogenes*, *Klebsiella oxytoca*, *Proteus mirabilis*, *S. aureus*, *Listeria ivanovii*, *Salmonella typhimurium*, and *P. aeruginosa*. The BS exhibited the strongest activity, with MIC values ranging from 0.05 to 2 μg/mL, while the 11S globulin displayed comparatively weaker effects. Scanning electron microscopy confirmed bacterial cell deformation and decay, indicating membrane disruption as a probable mechanism of action [176].

Moreover, the methylated derivative of 11S globulin (M11S) exhibited even greater antibacterial and anti-biofilm activity. Compared to 11S, M11S displayed significantly lower MIC values (0.025–0.50 μg/mL), surpassing some antibiotics by 40–80 times against Gram-positive bacteria and 2–60 times against Gram-negative bacteria. Additionally, M11S was highly effective in inhibiting biofilm formation, reducing it by 85–94% in Gram-positive and 43–50% in Gram-negative bacteria [177]. These findings suggest that Lupinus-derived proteins and their derivatives could have valuable applications in both medical and industrial antimicrobial strategies.

One of the most potent antimicrobial proteins from *Lupinus* is BLAD (Banda de Lupinus alba doce), a 20 kDa polypeptide derived from *L. albus*. BLAD, has demonstrated strong antibacterial and antifungal properties, effectively inhibiting *P. aeruginosa* and *B. subtilis*. Its applications extend beyond pharmaceuticals, including its use as a natural food preservative, disinfectant, and cleaning agent. Additionally, when combined with chelating agents such as EDTA, BLAD exhibits enhanced antimicrobial synergy, further broadening its potential use in healthcare and food safety [178].

Recent studies have explored the use of Lupinus-based extracts for synthesizing antimicrobial nanoparticles. Zinc oxide (ZnO) and magnesium-doped ZnO (Mg-doped ZnO) nanoparticles synthesized using *L. albus* leaf extract exhibited strong antibacterial activity, particularly against *E. coli* and *S. typhi*, with inhibition zones of 24 mm and 22 mm, respectively. While ZnO also displayed antimicrobial potential, the Mg-doped variant demonstrated superior efficacy, highlighting its potential as an alternative antimicrobial strategy against drug-resistant pathogens [179].

While some *Lupinus* extracts directly inhibit bacterial growth, others modulate bacterial adhesion and biofilm formation. *L. mutabilis* extract, for instance, did not exhibit direct antibacterial activity against *E. coli* and *K. pneumoniae*. However, it significantly reduced bacterial adhesion to bladder epithelial cells under normal and high-glucose conditions, as well as inhibit *E. coli* biofilm formation—although this effect was only significant at normal glucose levels. These effects were attributed to the downregulation of uroplakin A, a protein that facilitates bacterial adhesion, and the upregulation of RNase, an antimicrobial peptide. Such findings suggest that *L. mutabilis* may serve as a dietary supplement for preventing urinary tract infections, particularly in diabetic patients at higher risk [180].

These studies underscore the broad antimicrobial potential of *Lupinus* spp., ranging from hydrolysates and alkaloids to proteins, peptides, and nanoparticles. These bioactive compounds offer promising applications in pharmaceuticals, food preservation, and biotechnology, making *Lupinus* an invaluable source of natural antimicrobial agents. Future research should focus on optimizing extraction methods, characterizing novel bioactive compounds, and assessing their efficacy in clinical and industrial settings.

### 9.2. Antioxidant Activity

*Lupinus* spp. are recognized for their rich composition of bioactive antioxidant compounds, including polyphenols, carotenoids, and tocopherols. These compounds play a fundamental role in neutralizing free radicals, thereby reducing oxidative stress and contributing to the prevention of chronic diseases such as cancer and inflammatory conditions [181]. Notably, the presence of free phenolics in lupin seeds ensures high bioaccessibility in the small intestine, enhancing their health benefits. Furthermore, processing methods influence antioxidant content, with proper techniques preserving or even enhancing these properties. Compared to many other crops, *Lupinus* exhibits superior antioxidant activity, making it an excellent candidate for developing nutritionally enriched food products [182].

Enzymatic hydrolysis has been widely applied to *Lupinus* proteins to enhance their antioxidant potential. Hydrolysates produced using enzymes such as pepsin, pancreatin, and flavourzyme demonstrated strong antioxidant activity, as evaluated through radical scavenging assays (DPPH, ABTS, OH) and iron chelation. Notably, certain peptide fractions exhibited IC_50_ values of 30 ± 5.3 µg/mL for iron chelation and 40 ± 3.9 µg/mL for hydroxyl radical scavenging, highlighting their potential for nutraceutical and functional food applications [183].

Similarly, *L. mutabilis* (tarwi), an Andean grain with a protein content higher than soybeans, was investigated for its antioxidant peptides. Hydrolysis with Alcalase for 60 min followed by Neutrase for 120 min produced bioactive peptide fractions with strong antioxidant properties. LC-MS/MS analysis identified 25 peptides, predicted to possess antioxidative activity and additional benefits such as ACE and dipeptidyl peptidase IV (DPP IV) inhibition, suggesting potential applications in hypertension and diabetes management [184]. Another study optimized hydrolysis conditions using alcalase (enzyme/substrate ratio of 1.87%, 138 min incubation), yielding hydrolysates with TEAC and ORAC values of 2.7 ± 0.1 and 3.8 ± 0.1 µmol Trolox equivalents/mg protein, respectively, demonstrating potent radical-scavenging activity [185].

Germination has been identified as a method to enhance the antioxidant properties of *L. angustifolius* sprouts due to increased synthesis of secondary metabolites. Germinated samples exhibited higher antioxidant activity compared to ungerminated ones, particularly after simulated gastrointestinal digestion (GID), which modified certain phenolic and flavonoid profiles while maintaining overall antioxidant capacity. Notably, 7-day germinated sprouts (G7) demonstrated effective inhibition of reactive oxygen species (ROS) production, reducing oxidative damage and reinforcing their health benefits [186].

Beyond germination, debittered *L. mutabilis* seeds retained significant antioxidant properties, even after processing. A study comparing 33 ecotypes to control varieties (*L. albus*, *L. angustifolius*, *L. luteus*) found that *L. mutabilis* exhibited significantly higher antioxidant capacity, as indicated by the ABTS assay, showing 40.2 mmol Trolox equivalent (TE)/kg dry matter (DM). These results correlated strongly with flavonoid and total phenolic compound concentrations, reinforcing the role of polyphenols in antioxidant activity [153].

The antioxidant activity of *Lupinus* spp. extends beyond food applications, with promising implications for disease management. Inflammatory bowel disease (IBD) is one condition where antioxidant peptides from *L. mutabilis* have shown potential. Encapsulated peptide fractions (UF3) retained over 80% of their antioxidant capacity post-processing into nanoparticles, with ionic gelling and spray freeze-drying yielding stable formulations. These nanoparticles maintained over 70% cell viability in colonic HT-29 cell lines, confirming their safety for therapeutic applications in oxidative stress-related conditions [187].

In diabetes research, *L. albus* seed ethanol extract demonstrated oxidative stress reduction in diabetic rats. Treatment led to decreased malondialdehyde (MDA) levels, indicating lower lipid peroxidation in liver and kidney tissues. Additionally, the extract enhanced the activity of antioxidant enzymes, including catalase (CAT), superoxide dismutase (SOD), glutathione peroxidase (GPx), and glutathione (GSH), reinforcing endogenous antioxidant defenses. These findings suggest that *L. albus* could serve as a natural therapeutic agent for managing oxidative stress in diabetic patients [188].

Given their potent antioxidant properties, *Lupinus*-derived compounds have considerable potential in functional food formulations. Hydrolysates from *L. mutabilis* flour, particularly those derived from raw flour hydrolyzed with pancreatin for 4 h, exhibited the highest antioxidant capacity. ORAC and FRAP assays confirmed enhanced bioactivity under these conditions, suggesting their suitability as natural antioxidants for food and health applications [172].

Additionally, antioxidant properties were retained in heat-treated *L. albus* seeds, indicating that cooking does not compromise their health benefits. Using DPPH and ABTS assays, the antioxidant capacity was measured, with EC_50_ values ranging from 3473.49 ± 201.98 to 3832.65 ± 235.87 (g sample/g DPPH), confirming their free radical scavenging ability. Notably, macromolecular polyphenols played a crucial role in sustaining this antioxidant activity, further reinforcing their dietary significance [158].

The antioxidant potential of *Lupinus* spp. is attributed to their rich composition of polyphenols, peptides, and bioactive compounds, which effectively combat oxidative stress. Enzymatic hydrolysis enhances their bioactivity, producing hydrolysates with strong radical-scavenging properties. Germination further increases antioxidant potency, while encapsulation techniques ensure peptide stability for disease-targeted applications. Studies highlight the relevance of *Lupinus* in managing oxidative stress-related diseases, such as IBD and diabetes, while its potential as a functional food ingredient offers further opportunities for nutritional innovation. These findings position *Lupinus* as a valuable natural source of antioxidants for both health and industrial applications.

### 9.3. Anticancer and Anti-Inflammatory Activity

*Lupinus*-derived bioactive compounds, including protein hydrolysates, peptides, and alkaloids, have demonstrated significant anti-inflammatory properties, positioning them as potential natural alternatives to synthetic anti-inflammatory agents. Research on *L. mutabilis* revealed that aqueous seed extracts significantly reduced edema in rats at high doses (4000 mg/kg body weight), likely due to their ability to inhibit pro-inflammatory cytokines such as tumor necrosis factor (TNF-α) and interleukin-6 (IL-6) [189]. Additionally, the extract demonstrated enzyme inhibition against phospholipase A2 and cyclooxygenase-2 (COX-2), suggesting potential applications in preventing chronic inflammatory diseases.

Lupin protein hydrolysates (LPHs) have also shown promise in modulating inflammation at the cellular level. Using a co-culture model of Caco-2 intestinal epithelial cells and THP-1-derived macrophages, LPHs were found to cross the intestinal barrier and exert systemic anti-inflammatory effects. They significantly reduced nitric oxide (NO) and ROS production while suppressing pro-inflammatory cytokine expression, reinforcing their potential use in functional foods or nutraceuticals for inflammatory disease management [190].

Similarly, studies on *L. arboreus* demonstrated anti-inflammatory effects using formaldehyde- and egg-albumin-induced rat paw edema models. The methanol extract, administered at 30 mg/kg and 60 mg/kg, significantly reduced inflammation, an effect attributed to the presence of flavonoids, saponins, and terpenes—compounds known for their anti-inflammatory properties [191].

Furthermore, a study on lupin protein concentrate (LPC) enriched with deflamin, an inhibitor of MMP-9 (an enzyme linked to inflammatory bowel diseases), showed that LPC significantly reduced colitis severity in a TNBS-induced colitis mouse model. A diet of LPC-enriched wheat cookies maintained its protective effects post-baking, reducing colitis-associated lesions and oxidative stress while lowering COX-2 and TNF-α expression. These findings suggest that LPC could serve as a functional food ingredient for managing inflammation in inflammatory bowel diseases [192].

The anticancer potential of *Lupinus* spp. has been demonstrated through multiple mechanisms, including selective cytotoxicity against cancer cells, apoptosis induction, and modulation of cancer-related enzymes. Cytotoxicity studies on *L. angustifolius*, *L. luteus*, and *L. albus* extracts revealed their effectiveness in inhibiting prostate, skin, liver, and colon cancer cell proliferation. Alkaloids such as angustifoline and lupanine exhibited dose-dependent cytotoxic effects, with methanol extracts from bitter *Lupinus* varieties showing stronger anticancer activity compared to sweet lupins. Additionally, phenolic compounds and oligosaccharides in lupin seeds displayed antimutagenic properties, preventing genetic mutations that contribute to cancer development [189].

Moreover, ethanolic extracts from *L. angustifolius* roots and shoots were tested against breast cancer cell lines MCF-7 (estrogen receptor-positive) and BT20 (estrogen receptor-negative). The root extract at 200 µg/mL demonstrated the strongest effects, reducing MCF-7 proliferation by 71.03% and BT20 by 99.72%. The shoot extract achieved reductions of 64.7% and 86.32%, respectively. IC_50_ values indicated greater potency of the shoot extract against MCF-7 cells (18.06 µg/mL) compared to the root extract (52.3 µg/mL). These findings suggest that *L. angustifolius* extracts may serve as promising natural anticancer agents [193].

Further supporting these findings, a study on nine Australian *Lupinus* cultivars assessed their effects on colon carcinoma cell lines HCT116 and HT29. Mandelup and Jurien cultivars exhibited the strongest inhibition of cell proliferation, with apoptosis induction confirmed via Annexin V-FITC assays. The mechanism involved increased ROS generation and caspase-3/7 activation, leading to cancer cell cycle arrest and apoptosis. Notably, heat processing significantly reduced the anticancer activity, suggesting that the bioactive compounds responsible for these effects are heat-sensitive [194].

Beyond their direct anticancer and anti-inflammatory effects, *Lupinus*-derived compounds exhibit multifunctional health benefits, including metabolic regulation. LDPs have been identified as potential chemo-protective agents, mitigating the adverse effects of cancer treatments. Their strong antioxidant and anti-inflammatory properties help counteract oxidative stress and chronic inflammation, both of which contribute to tumor development and progression [163].

Additionally, LDPs have shown potential in regulating the glucose and lipid metabolisms—pathways frequently dysregulated in cancer. This suggests an indirect role in cancer prevention by improving metabolic health. Studies using ultraperformance liquid chromatography–tandem mass spectrometry (UPLC-MS/MS) have identified LDPs with anti-inflammatory activity, specifically through the modulation of the p38 MAPK signaling pathway in RAW 264.7 cells. By inhibiting p38 MAPK activation, these peptides reduced pro-inflammatory cytokine production, reinforcing their therapeutic potential for inflammation-related conditions [195].

The growing body of research on *Lupinus*-derived compounds highlights their promising role in both cancer prevention and inflammation management. Protein hydrolysates, bioactive peptides, alkaloids, and phenolic compounds contribute to their diverse bioactivities, ranging from selective cytotoxicity against cancer cells to the modulation of inflammatory pathways. The ability of lupin-derived peptides to cross the intestinal barrier, combined with their regulatory effects on oxidative stress and metabolic pathways, further strengthens their potential as functional food ingredients or nutraceuticals. 

Given these findings, *Lupinus* spp.—particularly bitter varieties—emerge as promising candidates for further research and potential application in natural anticancer and anti-inflammatory therapies.

### 9.4. Additional Health Benefits: Metabolic, Cardiovascular, Immune, and Digestive Health

*Lupinus* spp., particularly *L. angustifolius*, has demonstrated significant metabolic benefits, including obesity reduction and improved insulin sensitivity. Extracts from *L. angustifolius* significantly reduced obesity in high-fructose-diet-induced Wistar rats (*p* < 0.001) and lowered inflammatory markers (IL-6, TGFβ, TNF-α) (*p* < 0.001), suggesting potent anti-inflammatory and metabolic regulatory effects. Additionally, these extracts improved hematological parameters and decreased insulin resistance (*p* < 0.001). Metabolic hormone analysis revealed a significant reduction in leptin (*p* < 0.001) and non-significant changes in adiponectin and ghrelin, further indicating a role in energy homeostasis and appetite regulation [196].

The high protein and fiber content of *L. albus* supports energy metabolism by stabilizing blood glucose levels and promoting satiety. Protein hydrolysates derived from *Lupinus* have been shown to lower total and LDL cholesterol, reducing cardiovascular disease risks. These effects stem from bioactive peptides generated during digestion, which regulate cholesterol and glucose metabolism [197].

LPHs exhibit potent hypocholesterolemic effects. In ApoE-/- mice fed a Western diet, LPH treatment significantly reduced liver cholesterol and triglyceride levels by inhibiting HMG-CoA reductase (HMGCoAR), a key enzyme in cholesterol synthesis. LPHs also downregulated LDL receptor (LDLR) expression while modulating PCSK9 signaling, a crucial pathway in cholesterol metabolism. These findings suggest that LPHs can be developed as functional food ingredients to manage hypercholesterolemia [198].

Furthermore, LPHs effectively mitigated metabolic-associated fatty liver disease (MAFLD) and abdominal adiposity in mice. Treatment significantly reduced liver steatosis and downregulated lipid-metabolism-related genes (CD36, LDLR), indicating a regulatory role in lipid uptake. Additionally, LPHs enhanced hepatic antioxidant capacity and reduced inflammation, reinforcing their protective effects against obesity-related liver dysfunction [199].

A clinical case study of a 43-year-old man with metabolic syndrome (MetS) demonstrated that a diet rich in *L. luteus* combined with therapeutic lifestyle changes led to significant improvements in metabolic parameters. After six months, body weight decreased by 26.85%, BMI by 26.95%, and blood pressure by 13%. Cholesterol levels showed remarkable reductions, including a 53.84% decrease in total cholesterol and a 57.84% reduction in LDL cholesterol. Additionally, liver function improved, with ALT levels dropping by 61.62%. These findings suggest that dietary interventions incorporating lupin proteins offer a viable alternative to pharmaceutical treatments for MetS management [200].

The conglutin gamma (Cγ) protein from *L. albus* has demonstrated antidiabetic properties in experimental models. In diabetic rats, Cγ treatment significantly reduced hyperglycemia, showing an effect comparable to glimepiride, a conventional diabetes medication. Additionally, Cγ enhanced insulin secretion and sensitivity while modulating adipocytokine levels (leptin, adiponectin), both crucial for metabolic regulation. It also improved lipid profiles and reduced biomarkers associated with metabolic syndrome (apelin, NAMPT, RBP4), highlighting its potential as a natural antidiabetic agent [201].

However, a clinical study on the effects of lupin-enriched foods in individuals with type 2 diabetes mellitus (T2DM) found no significant changes in glycemic control or blood pressure over two 8-week treatment periods. While participants maintained stable glucose levels and blood pressure, the study suggests that future research should focus on individuals with poorly controlled diabetes to assess whether lupin-based interventions can provide greater benefits [202].

LPHs also regulate immune responses by modulating cholesterol metabolism, a crucial factor in inflammatory processes. By decreasing LDLR and PCSK9 expression, LPHs indirectly influence inflammatory pathways, contributing to their overall health benefits [198].

Lupin proteins and fibers contribute to improved gut health. Their high fiber content promotes beneficial gut microbiota composition, aiding digestion and regulating blood sugar levels. Furthermore, *L. angustifolius* protein hydrolysates have shown protective effects against diet-induced metabolic disorders by reducing liver inflammation and enhancing antioxidant defenses, suggesting that lupin-based interventions could support long-term gut and metabolic health [199].

The bioactive compounds in *Lupinus* spp., particularly proteins and their hydrolysates, offer a range of health benefits spanning metabolic, cardiovascular, immune, and digestive health. Their cholesterol-lowering, anti-inflammatory, and glucose-regulating properties make them promising candidates for functional foods aimed at preventing and managing metabolic disorders. While further research is necessary, particularly in human clinical trials, the current findings position *Lupinus* as a valuable dietary component for promoting long-term health and disease prevention.

As presented in Table 3, Lupinus-derived compounds exhibit diverse bioactivities including anti-inflammatory, antidiabetic, hypocholesterolemic, anticancer, and antimicrobial effects. Protein hydrolysates, extracts, and specific peptides from lupin seeds effectively modulate immune responses, reduce oxidative stress, and improve the lipid and glucose metabolisms across various experimental models.

## 10. Potential Therapeutic Use of the Andean Plants Based on Clinical Trials

Although, in the four main databases that capture clinical trials worldwide, ClinicalTrials.gov (US), EU Clinical Trials Register (Europe), ANZCTR (Australia and New Zealand), and Clinical Trials Registry—India, there is an important number of trials that have already been completed or that have been conducted with most of the active ingredients deeply described in the previous sections; it is almost impossible to extrapolate these to the consumption of the whole product. For this reason, in this section, we analyze the few clinical trials accomplished with that premise, i.e., using the whole product.

### 10.1. C. quinoa

We found nine clinical trials. Of those, seven were intended to evaluate the efficacy of quinoa administration (in different formulations) on the glucose metabolism or lipid parameters, and two focused on the gut microbiome.

To note, quinoa consumption was significantly more effective than a multiple whole-grain diet in controlling impaired glucose tolerance by reducing postprandial glucose and HbA1c levels, improving insulin resistance, and enhancing lipid profiles [210]. Similarly, 2 h postprandial blood glucose, glycosylated hemoglobin, insulin resistance index, body mass index, and mean diastolic blood pressure in the quinoa group were statistically significantly lower than in the control group, while high-density lipoprotein cholesterol is higher [211].

Although daily consumption of quinoa, in the short term, tended to lower glucose, it had minimal effects on other cardiovascular disease risk biomarkers [212]. However, processed quinoa intake during 28 days decreases body mass index and glycosylated hemoglobin A1c (HbA1c) levels, maintains fasting plasma glucose levels, and increases the satiation and fullness (complete) degree in prediabetic patients [213]. Similarly, it has been found that quinoa flakes produced significant reductions in serum triglyceride, thiobarbituric acid reactive substances, vitamin E concentrations, and a reduction in total cholesterol and LDL-cholesterol, as well as an increase in reduced glutation, showing a possible beneficial effect of quinoa intake [214].

Moreover, small but favorable changes in body weight, body mass index, and circulating cholesterol concentrations were reported after consumption of quinoa, all of which may contribute to lowered cardiovascular disease risk in older adults [215]. Also, the consumption of 50 g quinoa/d lowers serum triglycerides in overweight and obese participants and reduces the prevalence of metabolic syndrome by 70% [216], and the rate of conversion to diabetes for participants in the quinoa group (7.8%) was significantly lower than in the control group (20.3%) [211].

The last two studies focused on the intestinal microbiome, showing that a quinoa-based cookie prevented imbalances in the gut microbiota composition derived from the administration of the high-fat-containing cookie [217]. However, substituting refined wheat flour with whole-grain quinoa flour was not able to significantly modulate the gut microbiome [218]. Overall, the bacterial community did not seem to be influenced by the bacterium or by the beverage, as expressed by the diversity indexes, but specific genera were affected, as reflected in changes in amplicon sequence variants. Consequently, Lactiplantibacillus plantarum P31891 showed potential to be categorized as a probiotic strain in the fermented quinoa-based beverage [219].

Further studies with higher levels of quinoa or longer exposure periods are needed to ascertain if there is a dose−response effect of quinoa and if these effects are able to translate into clinical outcomes.

### 10.2. Amaranth

We found eight clinical trials, focused on three main aspects: glucose and lipids, iron deficiency anemia, and oxidative stress.

Compared to rapeseed oil, amaranth oil had a slight positive effect on adiponectin levels, but negatively affected ApoB concentrations and ApoB/A1 ratio. There were no differences in tumor necrosis factor-alpha, oxidized low-density lipoprotein, apolipoproteins E, and glucose and insulin homeostasis markers [220]. Similarly, a positive influence of press non-raffinate amaranth and sunflower oils has been shown, as well as their mix on carbohydrates and lipids metabolism exchange on patients with diabetes mellitus 2 types [221].

The supplementation of amaranth seed oil or rapeseed oil caused a significant reduction in weight, body mass index, waist circumference, hip circumference, fat mass, and visceral fat mass (VFM) in both groups. Except for HOMA-IR, there were no statistical differences between the clinical parameters of all groups. However, a trend toward improved insulin levels and HDL% was also noticeable [222]. Also, amaranth grain contains tocotrienols and squalene compounds, which are known to affect cholesterol biosynthesis. Based on these properties of amaranth oil, we hypothesize that it could be of significant benefit for patients with cardiovascular disease [149]. However, the use of amaranth oil instead of rapeseed oil may increase cardiovascular risk in obese and overweight subjects [223]. 

The prevalence of anemia was significantly lower with processed amaranth bread (32%) compared to the maize group (56%). The risk of iron deficiency anemia is significantly lower in the amaranth group, but there was no difference in iron deficiency [224].

Elevating plasma nitrites through dietary nitrates (NO_3_^−^) supplementation is associated with enhanced muscle efficiency, fatigue resistance, and performance. Recent evidence identified an alternative source rich with dietary NO_3_^−^ as a possible nitric oxide precursor, amaranth (*A. hypochondriacus*), with a standardized concentration of 9–11% of NO_3_^−^. Long-term (6 days) use of dietary NO_3_^−^ from amaranth may improve the aerobic capacity during ICE in young physically active male persons. It can be recommended as a nutritional supplement during the last week of preparation for competition in endurance events [225]. Also, a single oral dose of amaranth extract increased the NO_3_^−^ and NO_2_^−^ levels in the body by at least 8 h. The increase in NO_3_^−^ and NO_2_^−^ levels can help to improve the overall performance of people involved in vigorous physical activities or sports [147].

Supplementation with amaranth oil caused mild pro-oxidant activity, resulting in the improved uptake of oxidative destruction products and modulation of catalase and SOD activity, with the subsequent development of an antioxidant effect [226].

### 10.3. Lupinus

This has been the most studied product and with several clinical trials, most of them focusing on metabolic and cardiovascular diseases. Below is a summary of this available evidence.

Initial evidence, almost 40 years ago, showed that *L. termis* extractive was effective in the treatment of chronic eczema, comparable to those obtained with the corticoid therapy [227]. Then, in the decade of the 1990s, the protein quality of *L. albus* cv Multolupa was evaluated in young adult males using the nitrogen balance technique at graded levels of N intake, and revealed that the net protein utilization of lupin was 77% that of egg protein [228]. Similarly, the absorption of Zn from lupin-protein foods was found to be higher than from comparable soya-bean products; therefore, lupin milk could be an attractive alternative to soya-bean milk for infant formulas [229].

The addition of Australian sweet lupin (*L. angustifolius*) kernel fiber (LKFibre) to the diet provided favorable changes to some serum lipid measures in men, which, combined with its high palatability, suggest that this novel ingredient may be useful in the dietary reduction in coronary heart disease risk [230]. However, the Australian sweet lupin (*L. angustifolius*) flour (ASLF) addition to the breakfast reduced its glycemic index and raised its insulinemic index, but did not affect palatability, satiety, or food intake [231]. Finally, the addition of LKFibre to the diet incorporated into food products stimulated colonic bifidobacteria growth, which suggests that this dietary fiber may be considered a prebiotic and may improve some markers of healthy bowel function and colon cancer risk in men [232].

Although it is not a therapeutic use, it is important to stress that, in peanut-sensitized patients, clinically relevant sensitization to either lupine, pea, or soy occurs frequently. The ED for lupine is low (0.5 mg), which is only fivefold higher than for peanuts. Also, lupine-enriched pasta can be tolerated by most subjects suffering from peanut allergy, but a sizeable minority (2/12 of them in this case) can develop potentially dangerous clinical reactions. Patients are not aware of the lupine allergy and the presence of lupine in food, indicating that education is important to build awareness [233].

In overweight and obese men and women who were assigned to replace 15–20% of their usual daily energy intake with lupin kernel flour-enriched bread, was shown that increasing protein and fiber in bread with lupin kernel flour may be a simple dietary approach to help reduce blood pressure and cardiovascular risk [234]. In the same sense, lupin protein, compared to casein, slightly lowered the concentration of LDL cholesterol in hypercholesterolemic subjects, without altering HDL cholesterol. No or minor effects of lupin protein were observed on circulating glucose, homocysteine, and plasma amino acids [235]. However, there is not enough support for the proposal that an ad libitum diet enriched in LKFluour resulted in moderate changes in both protein and fiber intakes and can benefit body weight and composition or fasting blood lipids, glucose, and insulin concentrations in overweight men and women with mildly elevated total cholesterol concentrations [236]. Moreover, although regular consumption of lupin-enriched bread can lower blood pressure, these results do not support for the hypothesis that this is via effects on oxidative stress or vascular function [237].

A diet higher in protein and fiber derived from lupin-enriched foods does not enhance weight loss or improve the maintenance of weight loss. However, such a diet may provide cardiovascular health benefits in terms of insulin sensitivity and blood pressure [238]. Also, adding lupin or soya to a carbohydrate-rich beverage reduces glycaemia acutely in type 2 diabetic individuals. Although glucose and C-peptide responses did not differ significantly between lupin and soya, lupin resulted in a lower insulin response compared to soya. This may have a beneficial role in glycemic management [239]. Similarly, it has been demonstrated the hypocholesterolemic activity and potential clinical benefits of consuming lupin protein on plasma total and LDL-cholesterol levels [240]. The role of raw *L. mutabilis* on blood glucose and insulin was evaluated in normoglycemic and dysglycemic subjects, showing that *lupinus* consumption could be a feasible and low cost alternative to treat chronic hyperglycemic diseases [240]. Further study indicates that consumption of cooked *L. mutabilis* or its purified alkaloids decreased blood glucose and insulin levels, while none of the volunteers in either group presented side effects [241].

In hypercholesterolemic subjects, lupine protein can beneficially modulate plasma LDL cholesterol at least over the short term [242], and later has been hypothesized that the blood lipid-lowering effects of LF may be mainly attributed to the formation of short-chain fatty acids, specifically propionate and acetate [243].

## 11. Concluding Remarks

The Andean region has long served as a reservoir of nutritionally and pharmacologically valuable plant species, many of which remain underexplored in contemporary research. As demonstrated in this review, quinoa, amaranth, and lupinus harbor a diverse array of bioactive molecules that exhibit potent antioxidant, antimicrobial, anti-inflammatory, and metabolic health benefits. These bioactive compounds, shaped by extreme environmental pressures, not only enhance the resilience of these plants, but also underscore their potential as functional food ingredients and therapeutic agents.

Despite their extensive traditional use, the full medicinal and nutritional potential of Andean plants remains largely untapped. Emerging research underscores their ability to modulate cellular responses, mitigate oxidative stress, and improve metabolic regulation, supporting their integration into personalized nutrition, preventive healthcare, and drug discovery. However, several critical challenges must be addressed to fully harness their benefits:Standardization and bioavailability: The optimization of extraction techniques, formulation strategies, and dosage recommendations is necessary to ensure consistent bioactivity and efficacy [244].Clinical validation: Rigorous clinical trials are essential to substantiate their therapeutic properties and establish evidence-based applications [245].Sustainability and conservation: Agricultural and conservation strategies must prioritize biodiversity preservation [246] and the safeguarding of indigenous knowledge while fostering equitable economic opportunities for local communities.

Moving forward, interdisciplinary collaborations among biochemists, nutritionists, pharmacologists, and agronomists will be instrumental in translating ethnobotanical knowledge into scientifically validated applications. The promotion of Andean plants as functional foods and nutraceuticals presents a valuable opportunity to address global nutrition challenges, enhance food security, and advance novel therapeutic strategies.

The development of new functional foods and nutraceuticals based on ancestral plants represents a significant opportunity for innovation in the food sector. These products not only offer health benefits, but also respond to the growing consumer demand for foods that contribute to overall well-being [247]. By integrating traditional wisdom with modern scientific methodologies, these plants can play a pivotal role in shaping sustainable health solutions for future generations.

### Considerations on Antinutritional Factors

While quinoa (*Chenopodium quinoa*), amaranth (*Amaranthus* spp.), and lupin (*Lupinus* spp.) are widely recognized for their rich repertoire of bioactive compounds (Section 4, Section 6, and Section 8) and associated health benefits (Section 5, Section 7, and Section 9), these crops also harbor antinutritional factors (ANFs) that can impair nutrient bioavailability and digestive efficiency if not properly mitigated.

In quinoa, saponins—predominantly located in the seed pericarp—represent the most abundant ANFs, with concentrations ranging from 0.2 to 2.4 g/100 g [19]. These compounds contribute to the characteristic bitterness of the seeds and can reduce protein digestibility. However, their levels can be substantially lowered through conventional processing techniques such as washing or dehulling, which also preserve their antimicrobial activity (Section 5.1). In addition, quinoa contains phytic acid (1.05–1.35 g/100 g) and tannins (0.023–0.031 g/100 g), both of which may chelate essential minerals and limit their absorption [19].

Amaranth seeds similarly contain phytates (0.5–0.61 g/100 g), tannins (0.043–0.116 g/100 g catechin equivalents), and trypsin inhibitors (0.94–5.46 TIU/mg protein), which are known to interfere with protein digestion and mineral uptake. Nonetheless [248], the levels of tannins in amaranth are moderate relative to other pseudocereals, and targeted processing strategies such as germination or dehulling—discussed in Section 6—have been shown to markedly reduce these ANFs, thereby enhancing the nutritional profile of the seeds (Section 7.4).

In the case of lupin, particularly *Lupinus mutabilis*, quinolizidine alkaloids such as lupanine (0.1–4 g/100 g in bitter varieties) constitute the primary antinutritional concern, given their potential toxicity. These alkaloids can be effectively removed through debittering techniques, including soaking and repeated washing [249]. Additionally, lupin contains moderate levels of phytates (0.3–0.9 g/100 g) and trace amounts of trypsin inhibitors. Importantly, sweet lupin cultivars, as highlighted in Section 8, exhibit minimal alkaloid content, significantly improving their safety and nutritional suitability [181,250]. As detailed in Section 9.4, these processing steps are consistent with the metabolic health benefits attributed to lupin consumption.

Across all three species, both traditional and contemporary processing methods—such as washing, cooking, fermentation, and germination—have proven to be effective in reducing ANF content, as supported by improved nutrient bioavailability in clinical settings (Section 10). Notably, certain ANFs, such as quinoa-derived saponins, may exert beneficial biological activities (e.g., antioxidant and anti-inflammatory effects) at low concentrations, thereby complementing the therapeutic properties reviewed herein. Ultimately, the strategic integration of optimized processing techniques and targeted cultivar selection, as proposed in Section 11, holds considerable promise for enhancing the nutritional and medicinal value of these emblematic Andean crops.

## Figures and Tables

**Figure 1 nutrients-17-01749-f001:**
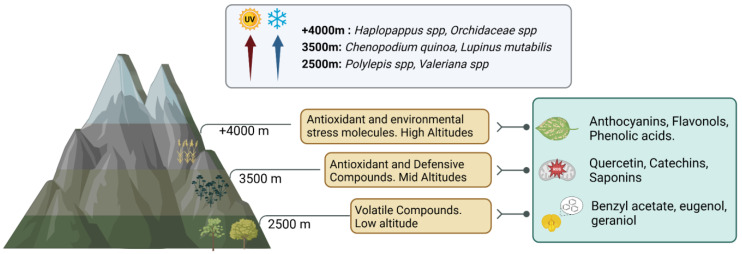
The Andean environment and its impact on phytochemistry.

**Figure 2 nutrients-17-01749-f002:**
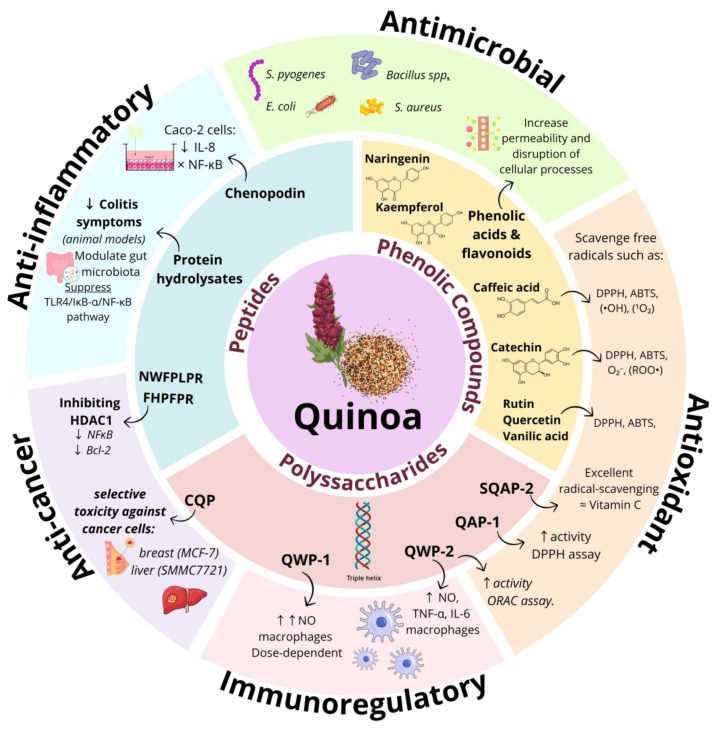
Quinoa’s compounds and their bioactivities. Quinoa has antimicrobial, antioxidant, anti-inflammatory, anticancer, and immunoregulatory effects. Its phenolic compounds and flavonoids scavenge free radicals (2,2-Diphenyl-1-picrylhydrazyl (DPPH) assay, 2,2′-azino-bis(3-ethylbenzothiazoline-6-sulfonic acid (ABTS) assays). Peptides and polysaccharides modulate inflammation (TLR4/NFκB pathway, TNF-α, IL-6). Its anticancer activity involves HDAC1 inhibition, NFκB modulation, and Bcl-2 regulation, promoting apoptosis.

**Figure 3 nutrients-17-01749-f003:**
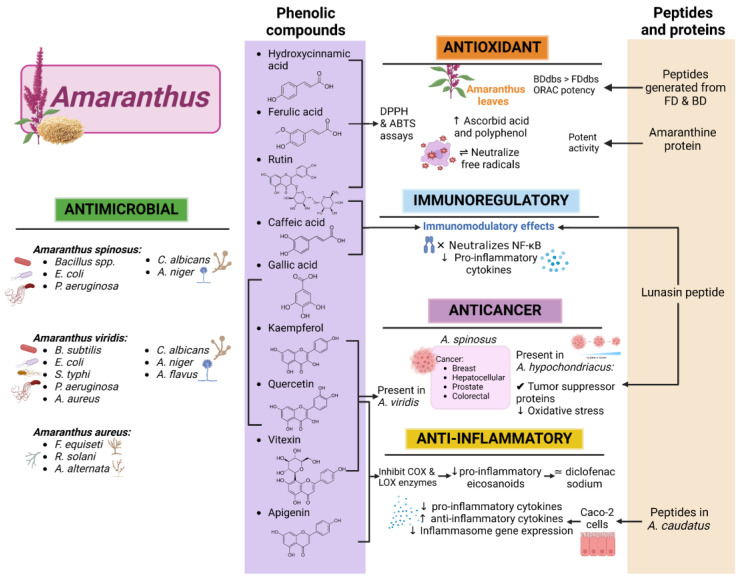
Amaranth’s bioactive compounds and their functions: Amaranth possesses antimicrobial, antioxidant, immunoregulatory, anti-inflammatory, and anticancer properties. Its phenolic compounds scavenge free radicals (DPPH, ABTS) and exhibit antimicrobial effects. Peptides and amaranthine protein enhance antioxidant activity, while lunasin and phenolics contribute to immunoregulation and target breast, prostate, colorectal, and liver cancer cells. Peptides and phenolics also exert anti-inflammatory effects by inhibiting lipoxygenase (LOX) and cyclooxygenase (COX) enzymes and modulating cytokines and eicosanoids.

**Figure 4 nutrients-17-01749-f004:**
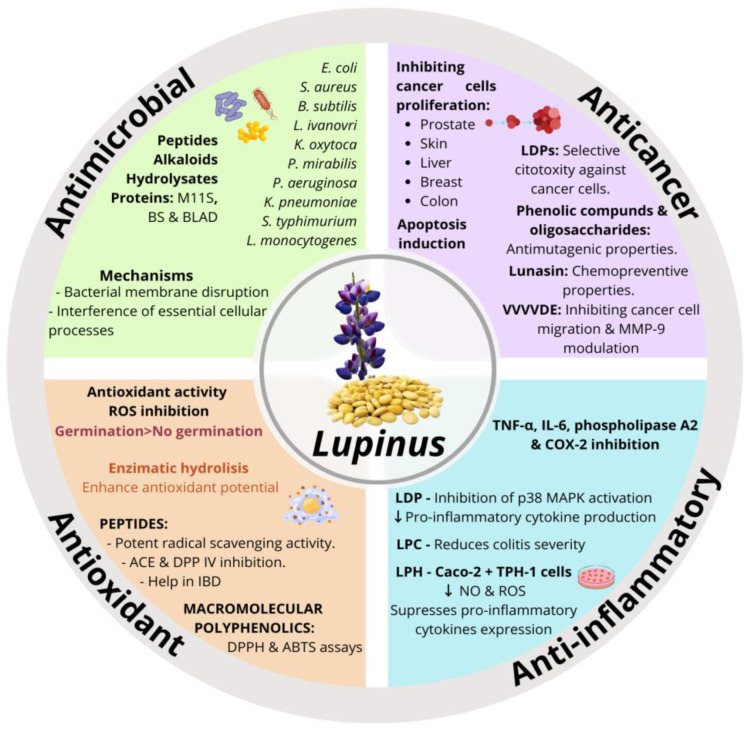
Bioactive properties of Lupinus: Lupinus exhibits antimicrobial, antioxidant, anti-inflammatory, and anticancer properties. Its peptides, alkaloids, and proteins disrupt bacterial membranes, while phenolics and oligosaccharides contribute to its anticancer effects by inducing selective cytotoxicity and apoptosis. Peptides and phenolics modulate cytokines and inhibit inflammatory molecules, while their antioxidant activity is enhanced during germination through improved radical scavenging capacity.

**Table 1 nutrients-17-01749-t001:** Role of quinoa in modulating cellular responses and disease progression.

Health Benefit	Origin	Evidence/Mechanism of Action	Model Used to Determination	References
Immunomodulatory	Protein hydrolysate and peptide fraction of *C. quinoa* cv. INTA Hornilloa	The protein hydrolysate elicited a balanced immunomodulatory response by increasing pro-inflammatory mediators (↑ IFN-γ, ↑ TNF-α) while regulating anti-inflammatory pathways (↑ IL-10, ↓ IL-6). The peptide fraction strongly stimulated phagocytosis, potentially boosting the innate immune response.	In vivo: BALB/c mice; peritoneal and splenic macrophages isolated from BALB/c mice.	[71]
Anti-inflammatory and Immunomodulatory	Protein-rich fraction (PRF) and PRF treated with food-grade enzymes from *C. quinoa*	The fractions reduced inflammation by inhibiting pro-inflammatory cytokines (↓ IL-6, ↓ TNF-α, ↓ IL-12p40, ↓ IL-27p28), chemokines (↓ MCP-1, ↓ MIP-1α), and nitric oxide (↓ NO) in M1 macrophages. They promoted M2 polarization by increasing IL-10 levels and arginase activity, and modulated the Toll-like receptor.	In vitro: Primary bone marrow-derived macrophages (BMDMs) and dendritic cells (BMDCs) from BALB/c mice; murine cell lines J774A.1 and JAWS II.	[58]
Hepatoprotective and Obesity prevention	Protein-rich fraction (PRF) and oil fraction from seeds of *C. quinoa*	The protein-rich fraction promoted monocyte migration to the liver, where they differentiated into monocyte-derived macrophages (MoMFs) that facilitated hepatic triglyceride mobilization. Innate lymphoid cells (ILC2s/ILC3s) were modulated, reducing hepatic triglyceride content and inflammation.	In vivo: Rag2^−^/^−^ mice (lacking mature T and B lymphocytes, but retaining innate immune cells) and Rag2^−^/^−^ IL2^−^/^−^ mice (further deficient in IL-2, affecting ILC survival and function).	[72]
Anti-obesity and Gut microbiota modulation	Quinoa peptides from seeds of *C. quinoa* Willd	Upregulation of hepatic PPAR-α (↑ FABP5, ↑ HMGCS1) to enhance lipid oxidation and downregulating PPAR-γ (↓ ANGPTL4, ↓ PCK1) to suppress adipogenesis. Reduced systemic inflammation (↓ TNF-α, ↓ IL-6, ↓ IL-1β) and restored gut microbiota diversity by normalizing the Firmicutes/Bacteroidetes ratio (↑ Muribaculaceae, ↓ Desulfobacterota).	In vivo: Murine obesity model in male C57BL/6 mice.	[73]
Hypolipidemic and Hepatoprotective	Husk of *C. quinoa* containing triterpenoid saponins	Saponins lowered serum total cholesterol, triglycerides, and low-density lipoprotein cholesterol while mitigating liver injury (↓ ALT, ↓ AST, ↓ total bile acids) and systemic inflammation (↓ TNF-α, ↓ IL-6). They enhanced hepatic fatty acid β-oxidation and bile acid metabolism, and modulated gut microbiota by decreasing the Firmicutes/Bacteroidetes ratio and increasing beneficial taxa.	In vivo: Male Sprague Dawley (SD) rats with induced hypercholesterolemia.	[74]
Hepatoprotective and Microbiota Modulation	*C. quinoa* Willd. whole grain	Hepatoprotective effects by reducing hepatic lipid accumulation (↓ NEFAs, ↓ TG, ↓ TC) and improving liver function (↓ ALT, ↓ ALP, ↑ A/G ratio) to mitigate steatosis. It also enhanced antioxidant defenses (↑ SOD, ↑ GSH-PX, ↓ MDA) and normalized systemic inflammation, while restoring intestinal microbiota homeostasis (↑ Akkermansia, ↑ Blautia; ↓ Clostridium, ↓ Turicibacter).	In vivo: Male Sprague Dawley (SD) rats fed a high-fat diet.	[75]
Polycystic ovary syndrome (PCOS) and Gut Microbiota Modulation.	Extract from dried and ground seeds of *C. quinoa* Willd	Quinoa supplementation improved PCOS by ameliorating hyperandrogenemia and enhancing hormonal profiles (↓ T, ↓ LH, ↓ LH/FSH, ↓ FINS, ↓ HOMA-IR; ↑ E2). It activated the PI3K/AKT/mTOR pathway (↑ PI3K, ↑ AKT, ↑ mTOR) and modulated autophagy markers (↑ Bcl-2, ↑ p62; ↓ Beclin1, ↓ULK1, ↓LC3B), while reinforcing the intestinal barrier.	In vivo: Letrozole-induced PCOS model in Sprague Dawley rats.	[76]
Gut microbiota modulation (Prebiotic Effects)	Polysaccharide extract from *C. quinoa* Willd	Quinoa polysaccharides exhibited prebiotic effects by modulating gut microbiota composition (↑ Bifidobacterium, ↑ Collinsella, ↑ Prevotella, ↑ Bacteroides; ↓ Clostridia) and increasing short-chain fatty acids (↑ butyrate, ↑ propionate), stimulating GLP-1 and PYY secretion.	In vitro: Fermentation model using human fecal microbiota.	[77]
Anticancer	Extract obtained from *C. quinoa* bran	The extract upregulated apoptotic biomarkers (↑ Caspase-3, -8, -9; ↑ Bax), decreased mitochondrial membrane potential, and downregulated the anti-apoptotic protein (↓ Bcl-2), leading to reduced cell proliferation and colony formation. Also reduced tumor volume and weight in DLD-1 xenograft nude mice with minimal systemic toxicity.	In vitro: DLD-1 and HCT-8 colorectal cancer cell lines. In vivo: Xenograft model in BALB/c nude mice.	[78]
	Terpenoid compounds obtained from *C. quinoa* bran	The extract down regulated drug resistance proteins (↓ P-gp, ↓ MRP1, ↓ BCRP) and upregulated miR-495-3p, which in turn decreased DNMTs (↓ DNMT1, ↓ DNMT3a, ↓ DNMT3b) expression. This reactivation enhanced chemotherapy efficacy (↑ 5-Fluorouracil (5-Fu) accumulation) and induced apoptosis (↑ Caspase-3, -8, -9).	In vitro: HCT-8 (5-Fu sensitive) and HCT-8/Fu (5-Fu resistant) colon cancer cell lines. In vivo: Xenograft in BALB/c nude mice.	[79]
Skin anti-aging	Leachate and fractions from seeds of *C. quinoa*	Quinoa leachate and its fractions demonstrated robust collagen protection via the downregulation of MMP-1 mRNA expression and inhibition of MMP-9 enzymatic activity, along with antioxidant effects (ROS scavenging). Additionally, it exhibited anti-pigmentation effects by suppressing tyrosinase activity (↓ melanin synthesis).	In vitro: Human dermal fibroblasts (HDF), murine macrophages (RAW 264.7), murine hepatocytes (H4IIE), and human colon adenocarcinoma cell line (Caco-2).	[80]

**Arrows explanation:** ↑ indicates upregulation or increase in the expression or activity of the corresponding molecule or parameter; ↓ indicates downregulation or decrease in the expression or activity of the corresponding molecule or parameter. **Abbreviations:** 5-Fluorouracil (5-Fu); Albumin/Globulin ratio (A/G ratio); protein kinase B (AKT); alkaline phosphatase (ALP); Alanine aminotransferase (ALT); Angiopoietin-like 4 (ANGPTL4); Aspartate aminotransferase (AST); Bagg albanian laboratory breed (BALB); breast cancer resistance protein (BCRP); pro-apoptotic protein Bcl-2-associated X protein (Bax); factor involved in autophagy (Beclin1); bone marrow–derived dendritic cells (BMDCs); anti-apoptotic protein (Bcl-2); bone marrow–derived macrophages (BMDMs); Cysteine-aspartic proteases involved in apoptosis (Caspase-3, -8, -9); DNA methyltransferases (DNMT1, DNMT3a, DNMT3b); Estradiol (E2); Fatty Acid Binding Protein 5 (FABP5); Follicle-stimulating hormone (FSH); fasting insulin or related insulin parameter (FINS); Glutathione peroxidase (GSH-PX); Glucagon-like peptide-1 (GLP-1); HCT-8 human colorectal cancer cell line (HCT-8); 3-Hydroxy-3-methylglutaryl-CoA synthase 1 (HMGCS1); Homeostatic Model Assessment for Insulin Resistance (HOMA-IR); IL-2 knockout murine model (IL2^−^/^−^); Interleukin 27 subunit p28 (IL-27p28); Interleukin 12 subunit p40 (IL-12p40); Interleukin 10 (IL-10); Interleukin 6 (IL-6); Interleukin 1 beta (IL-1β); Interferon gamma (IFN-γ); J774A.1 murine macrophage cell line (J774A.1); JAWS II murine immature dendritic cell line (JAWS II); innate lymphoid cells type 2 and type 3 (ILC2s/ILC3s); autophagy-related protein microtubule-associated protein 1A/1B light chain 3B (LC3B); Luteinizing Hormone (LH); Luteinizing Hormone to Follicle-Stimulating Hormone ratio (LH/FSH); M1 macrophages, pro-inflammatory type (M1); M2 macrophages, pro-inflammatory type (M2); Monocyte Chemoattractant Protein-1 (MCP-1); Malondialdehyde (MDA); Macrophage Inflammatory Protein-1α (MIP-1α); microRNA 495-3p (miR-495-3p); Matrix Metalloproteinase 9 (MMP-9); Monocyte-derived macrophages (MoMFs); Multidrug Resistance-Associated Protein 1 (MRP1); mammalian target of rapamycin (mTOR); non-esterified fatty acids (NEFAs); Nitric Oxide (NO); Sequestosome 1 (p62); Polycystic Ovary Syndrome (PCOS); Phosphoenolpyruvate carboxykinase 1 (PCK1); Phosphoinositide 3-kinase (PI3K); P-glycoprotein (P-gp); Peroxisome proliferator-activated receptor alpha (PPAR-α); Peroxisome proliferator-activated receptor gamma (PPAR-γ); protein-rich fraction (PRF); Peptide YY (PYY); Rag2 knockout murine model (Rag2^−^/^−^); reactive oxygen species (ROS); superoxide dismutase (SOD); Testosterone (T); Triglycerides (TG); Tumor Necrosis Factor alpha (TNF-α), total cholesterol (TC); Unc-51 like autophagy activating kinase 1 (ULK1).

**Table 2 nutrients-17-01749-t002:** Role of amaranth in modulating cellular responses and disease progression.

Health Benefit	Origin	Evidence/Mechanism of Action	Model Used to Determination	References
Anti-inflammatory	Extract from the leaves of *A. hybridus* cv. Kongei and cv. IP-7.	The extract inhibited nuclear factor kappa B (NF-κB) signaling in macrophages, thereby reducing pro-inflammatory cytokine expression (↓ TNF-α, ↓ IL-6). The active compound was identified as 2-caffeoylisocitric acid.	In vitro: Murine macrophage cell line RAW 264.7.	[124]
	Protein hydrolysates from *A. caudatus* L.	Protein hydrolysates modulated the NOD-like receptor family pyrin domain containing 3 (NLRP3) inflammasome pathway, reduced intestinal inflammation, and increased anti-inflammatory cytokines.	In vitro: Human colon adenocarcinoma cell line (Caco-2).	[125]
	Extract from the leaves of *A. lividus* and *A. tricolor*.	The extract downregulated pro-inflammatory cytokines (↓ TNF-α, ↓ IL-1, ↓ IL-6) in neuronal cells, thereby reducing oxidative stress and neuroinflammation induced by advanced glycation end products.	In vitro: Human neuroblastoma cell line (SH-SY5Y).	[126]
Immunomodulatory	Boiled leaves of *A. cruentus*, *A. viridis*, and *A. hybridus*.	Food supplementation with boiled leaves restored the balance between Th1/Th2 and Th17/Treg responses by enhancing lymphocyte activation, increasing IL-4 and IL-10 secretion, and reducing IFN-γ and IL-17 levels.	In vivo: Female Wistar rats immunized with type II collagen.	[127]
Antidiabetic	Ethanolic hydroalcoholic extract from seeds of *A. caudatus*.	The extract improved glycemic control by increasing insulin production in pancreatic islets through the activation of L-type calcium channels, PKA, PKC, and G protein-coupled exocytosis, resulting in better glucose tolerance and reduced HbA1c levels.	In vivo: Type 2 diabetic Goto-Kakizaki rats and healthy Wistar rats. In vitro: Pancreatic islets from both models.	[128]
	Aqueous extract of fresh leaves of *A. tricolor*.	The extract lowered blood glucose levels, improved lipid profiles (↓ cholesterol, ↓ LDL, ↑ HDL), and protected pancreatic β-cells against oxidative stress.	In vivo: Albino Wistar rats with alloxan-induced diabetes.	[129]
Hepatoprotective	Ethanolic extract of whole plant of *A. spinosus*	The extract neutralized trichloromethyl radicals (CCl_3_^+^), reduced oxidative damage markers (↓ MDA, ↓ lipid hydroperoxides), and increased antioxidants (↑ GSH, ↑ SOD, ↑ CAT). It also normalized liver enzymes (↓ AST, ↓ ALT, ↓ ALP).	In vivo: Sprague Dawley rats intoxicated with carbon tetrachloride.	[130]
	Ethanolic extract of leaves *A. tricolor*	The extract scavenged free radicals, inhibited lipid peroxidation (↓ MDA), increased endogenous antioxidants (↑ NP-SH, ↑ total proteins), normalized liver enzymes (↓ GOT, ↓ GPT), and reduced necrosis and inflammation. It preserved metabolic enzymatic activity, evidenced by decreasing pentobarbital-induced sleep time.	In vivo: Wistar rats intoxicated with carbon tetrachloride.	[131]
Anticancer	Protein hydrolysates from seeds of *A. caudatus*.	The hydrolysates induced apoptosis, evidenced by DNA fragmentation indicating nuclear condensation and chromatin degradation, caspase-3 activation, loss of membrane integrity, and phosphatidylserine translocation, and inhibited cell migration and viability in a breast adenocarcinoma cell line.	In vitro: Human breast adenocarcinoma cell line (MDA-MB-231).	[132]
	Methanol and aqueous extract form *A. hypochondriacus*, *A. caudatus*, *A. cruentus*, and *A. spinosus.*	The extract demonstrated greater anticancer activity by decreasing cell proliferation (↓ cell viability in an MTT assay). Notably, the aqueous extract from *A. hypochondriacus* showed higher cytotoxicity compared to the other species.	In vitro: Small-cell lung cancer cell line H69V, hepatocarcinoma cell line HepG2/C3A, and non-cancerous Vero cell line.	[133]

**Arrows explanation:** ↑ indicates upregulation or increase in the expression or activity of the corresponding molecule or parameter; ↓ indicates downregulation or decrease in the expression or activity of the corresponding molecule or parameter. **Abbreviations:** Alanine aminotransferase (ALT); Alkaline phosphatase (ALP); Aspartate aminotransferase (AST); Catalase (CAT); glutamic-oxaloacetic transaminase (GOT); glutamic-pyruvic transaminase (GPT); Glutathione (GSH); H69V small-cell lung cancer cell line (H69V); high-density lipoprotein (HDL); glycated hemoglobin (HbA1c); Hepatocellular carcinoma cell subclone (HepG2/C3A); Interferon gamma (IFN-γ); Interleukin 1 (IL-1); Interleukin 4 (IL-4); Interleukin 6 (IL-6); Interleukin 10 (IL-10); Interleukin 17 (IL-17); low-density lipoprotein (LDL); Malondialdehyde (MDA); colorimetric cell viability assay (MTT assay) (MTT); NOD-like receptor family, pyrin domain containing 3 (NLRP3); non-protein thiol groups (NP-SH); Protein Kinase A (PKA); Protein Kinase C (PKC); human neuroblastoma cell line (SH-SY5Y); superoxide dismutase (SOD); regulatory T cells (Treg); T helper type 1 cells (Th1); T helper type 17 cells (Th17); T helper type 2 cells (Th2); tumor necrosis factor alpha (TNF-α); African green monkey kidney cell line (Vero).

**Table 3 nutrients-17-01749-t003:** Role of *Lupinus* in modulating cellular responses and disease progression.

Health Benefit	Origin	Evidence/Mechanism of Action	Model Used to Determination	References
Anti-inflammatory	Protein hydrolysates from seeds of *L. angustifolius*	Inhibits the activation of the Th1 pathway (↓ IL-2, ↓ IL-12, ↓ IFN-γ, ↓ TNF), reduced the pro-inflammatory mediators of the Th17 (↓ IL-17) and Th9 (↓ IL-9) pathways to limit inflammation, and favored an anti-inflammatory profile through a relative increase in IL-4, IL-13, and IL-10 (↑ IL-4/IL-12, ↑ IL-13/IFN-γ, and ↑ IL-10/IFN-γ ratios).	In vitro: Human peripheral blood mononuclear cells (PBMCs).	[203]
	Protein hydrolysates derived from Lupin flour.	Lupinus peptides crossed the in vitro intestinal barrier in a co-culture system of Caco-2 cells (intestinal epithelium model) and THP-1 cells (immune response model), resulting in reduced mRNA expression and secretion of pro-inflammatory cytokines (↓ TNF-α, ↓ IL-1β, ↓ IL-6) and decreased oxidative stress markers (↓ ROS, ↓ nitrites).	In vitro: Human colon adenocarcinoma cell line (Caco-2) and THP-1 macrophages.	[190]
	Protein concentrate from seeds of *L. albus cv* Amiga	The lupin protein concentrates reduced gelatinolytic activity (↓ MMP-9) to limit tissue breakdown and downregulated pro-inflammatory mediators (↓ TNF-α, ↓ COX-2) to reduce mucosal damage while enhancing antioxidant defenses (↑ SOD, ↑ GPx) and reducing lipid peroxidation (↓ LPO) to alleviate oxidative stress.	In vitro: Human colon adenocarcinoma cell line (HT29). In vivo: CD-1 mice (TNBS-induced colitis) and Wistar rats (acetic acid-induced colitis).	[192]
Anti-diabetic	Ethanolic extract from seeds of *L. mutabilis*	Enhanced insulin secretion from pancreatic islets by modulating ion channels and signaling pathways (↑ action potential of L-type calcium channels, ↑ activity of PKA, PKC, and ↑ G protein-dependent exocytosis). Improvement in glycemic control as decreasing fasting glucose, reducing HbA1c.	In vivo: Goto-Kakizaki (GK) type 2 diabetic rats and healthy Wistar rats. In vitro: Pancreatic islets isolated from both models.	[204]
	Protein extracts from seeds of *L. albus*	The extracts reduced hepatic lipogenesis by downregulating SREBP-1c (↓ FAS, ↓ SCD1, ↓ GPAT) to suppress hepatic lipogenesis and upregulated enzymes involved in triglyceride hydrolysis (↑ LPL, ↑ HL, ↑ apoA5), thereby enhancing LDL clearance without significantly affecting PPAR-α–dependent genes (ACO, CPT-1a, CYP4A1) or insulin and glucagon concentrations.	In vivo: Sprague Dawley (SD) rats with induced hypercholesterolemia.	[205]
	Peptides P5 (LILPKHSDAD) and its derivative P5-met (LPKHSDAD) from lupin protein.	P5 and P5-met displayed hypocholesterolemic effects by reducing factors that promote LDLR degradation (↓ PCSK9, ↓ HNF-1α), increasing SREBP-2 and LDLR expression, and lowering the activity of HMGCoAR (a key enzyme in cholesterol synthesis), thereby enhancing LDL clearance (↑ LDL uptake).	In vitro: Human colon adenocarcinoma cell line (Caco-2) and human hepatocarcinoma cell line (HepG2).	[206]
	Peptide (GPETAFLR) from seeds of *L. angustifolius* L.	The GPETAFLR exhibited anti-inflammatory effects by reducing the production of pro-inflammatory mediators (↓ TNF-α, ↓ IL-1β, ↓ IL-6), as well as CCR2/CCL2 expression and the proportion of classical monocytes (CD14++CD16−), while increasing IL-10, favoring the presence of non-classical monocytes (CD14+CD16++).	In vitro: Primary human monocytes isolated from healthy donors.	[207]
Anticancer and anti-inflammatory	Ethanolic extracts (germinated and non-germinated) from seeds of *L. albus* and *L. angustifolius*.	Germinated seed extracts demonstrated greater anticancer and anti-inflammatory potential because of their increased content of genistein and cinnamic acid derivatives (↑ genistein, ↑ ferulic, ↑ rosmarinic, and ↑ caffeic) compared to non-germinated seeds.	In vitro: Breast cancer cell lines (MCF7, MDA-MB-231), ovarian cancer cell line (A2780), cervical cancer cell line (SiHa). In vivo: Ear inflammation model in female SKH-1 mice.	[208]
Anticancer	Isoforms of β-conglutin from seeds of *L. angustifolius* L.	The β-conglutins isoforms exhibited anticancer activity by decreasing cell proliferation, reducing oxidative stress (↓ ROS) and genotoxic damage (↓ γH2Ax), and inducing caspase-independent apoptosis (↓ caspase 3 activation). They regulated the SIRT1/FoxO1 pathway differently depending on the cell subtype: in MDA-MB-231 (p53 gain-of-function mutant), they promoted autophagy (↑ LC3B, ↓ p62), whereas in MCF-7 and SK-BR-3, they blocked this route and inactivate tumor stem cell traits.	In vitro Breast cancer cell lines MCF-7 (wild-type p53), SK-BR-3 (HER2+, nonfunctional p53), MDA-MB-231 (triple-negative, mutated p53), and non-cancerous mammary epithelial cell line (MCF-10A).	[209]
Antibacterial Activity	Extract from seeds of *L. mutabilis*	The extract exhibited anti-infective effects by modulating the host response, decreasing the expression of uroplakin1a (a key component of the uroepithelial cell surface that serves as a receptor for bacterial adhesion). Simultaneously, it increased the expression of the antimicrobial peptide RNase 7, a crucial component of the innate immune response in the urinary tract. Additionally, it inhibited biofilm formation.	In vitro: Human uroepithelial cell lines T24 and 5637 under normoglycemic and hyperglycemic conditions.	[180]

**Arrows explanation:** ↑ indicates upregulation or increase in the expression or activity of the corresponding molecule or parameter; ↓ indicates downregulation or decrease in the expression or activity of the corresponding molecule or parameter. **Abbreviations:** Urothelial cancer cell line (5637); A2780 human ovarian cancer cell line (A2780); Acyl-CoA oxidase (ACO); Apolipoprotein A5 (apoA5); receptor/chemokine involved in monocyte migration (CCR2/CCL2); colon adenocarcinoma cell line (Caco-2); CD-1 mouse strain (CD-1); phenotypic markers of classical monocytes (CD14++CD16−); phenotypic markers of non-classical monocytes (CD14+CD16++); Cyclooxygenase 2 (COX-2); Carnitine palmitoyltransferase 1a (CPT-1a); Cytochrome P450 4A1 (CYP4A1); fatty acid synthase (FAS); Forkhead box O1 (FoxO1); Glycerol-3-phosphate acyltransferase (GPAT); peptide sequence (GPETAFLR); Glutathione peroxidase (GPx); G protein (guanine nucleotide-binding) (G protein); Hemoglobin A1c (HbA1c); Human Epidermal Growth Factor Receptor 2 Positive (HER2+); Hepatic lipase (HL); 3-Hydroxy-3-methylglutaryl-CoA reductase (HMGCoAR); Hepatocyte nuclear factor 1 alpha (HNF-1α); colon adenocarcinoma cell line (HT29); Interferon gamma (IFN-γ); Interleukin 1 beta (IL-1β); Interleukin 10 (IL-10); Interleukin 12 (IL-12); Interleukin 13 (IL-13); Interleukin 17 (IL-17); Interleukin 2 (IL-2); Interleukin 4 (IL-4); Interleukin 9 (IL-9); L-type calcium channels (L-type); low-density lipoprotein (LDL); low-density lipoprotein receptor (LDLR); autophagy-associated protein LC3B (LC3B); lipoprotein lipase (LPL); lipid peroxidation (LPO); non-cancerous mammary epithelial cell line (MCF-10A); breast cancer cell line (MCF-7); triple-negative breast cancer cell line (MDA-MB-231); Matrix Metalloproteinase-9 (MMP-9); protein kinase A (PKA); protein kinase C (PKC); peptide sequence: LILPKHSDAD (P5); modified derivative of P5 (LPKHSDAD) (P5-met); proprotein convertase subtilisin/kexin type 9 (PCSK9); Peroxisome proliferator-activated receptor alpha (PPAR-α); tumor suppressor protein (p53); autophagy adaptor protein (p62); Ribonuclease 7 (RNase 7); Acyl-CoA desaturase 1 (SCD1); Sterol regulatory element-binding protein 1c (SREBP-1c); Sterol regulatory element-binding protein 2 (SREBP-2); cervical cancer cell line (SiHa); Sirtuin 1 (SIRT1); SKH-1 hairless mouse strain (SKH-1); breast cancer cell line (HER2+ and nonfunctional p53) (SK-BR-3); reactive oxygen species (ROS); superoxide dismutase (SOD); urothelial cancer cell line (T24); THP-1 human monocytic cell line (THP-1); Tumor Necrosis Factor (TNF); Tumor Necrosis Factor alpha (TNF-α); phosphorylated variant of histone H2A (DNA damage marker) (γH2Ax).

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
