# Peer review of "Recent Advances in the Therapeutic Potential of Bioactive Molecules from Plants of Andean Origin"

_nutrients, 2025, doi:10.3390/nu17111749_

Round 1
Reviewer 1 Report
Comments and Suggestions for Authors
Peer-Review Report
Journal: Nutrients
Manuscript ID: nutrients-3524514
Manuscript Title: Andean Plants: A Source of Bioactive Molecules for Modern Therapeutic Approaches – A Comprehensive Review
The title of the manuscript—“Andean plants as a source of bioactive molecules for modern therapeutic approaches”—suggests a specific focus on bioactive compounds derived from Andean plant species and their potential therapeutic applications. However, the content of the article does not sufficiently support this focus. Only limited attention is given to the actual properties of bioactive molecules specifically from Andean varieties of quinoa, amaranth, and lupin.
In particular:
- The majority of the information presented is general and qualitative, often repeating well-known data about bioactive compounds found in these crops, with little to no focus on their Andean origin. For instance in sections 2 and 3, the authors describe quinoa, amaranth, and lupin as historically important crops with origins in the Andes, and they discuss the influence of the Andean environment on the phytochemical properties of these plants. However, the article includes very little—if any—research specifically focused on these plant species grown in the Andean region.
- No quantitative data or comparative analysis is provided regarding bioactive profiles of Andean versus non-Andean plant varieties.
- The manuscript does not include any original studies or in-depth discussion of therapeutic effects, despite this being explicitly mentioned in the title. Potential therapeutic effects are not the same as actual therapeutic effects.
- The repetition of well-known information across multiple sections further weakens the overall impact. For example, the information presented in lines 599 to 605 consists of general chemical facts that are not specific to the extraction of compounds from quinoa, amaranth, or lupin. In this section, I would expect not only a qualitative description but also a quantitative comparison of the most effective methods for the extraction of phenolic compounds from these plants, especially in relation to Andean varieties.
- In section 5.2 author describe to general about antioxidants. For the
- In Sections 4 and 6, the authors describe specific groups of bioactive compounds found in quinoa and amaranth, including their chemical structures and health-related effects. However, this content is repeated in Sections 5 and 7, resulting in unnecessary redundancy. To improve clarity and coherence, I recommend consolidating and restructuring these sections.
In contrast, the section on lupin is the most clearly structured and informative part of the manuscript. It provides a more focused and coherent overview, which could serve as a model for improving the presentation of content in the other sections.
- Figures are not referenced or contextualized in the main text, reducing their contribution to the overall narrative.
- Authors should provide a clear description of the methodology used to select and review the literature, including databases searched, keywords, and inclusion/exclusion criteria.
- Furthermore, while the manuscript seems to aim for a comprehensive review, it lacks the critical depth and synthesis expected of such an article. The text often reiterates generally known information
Given the significant gap between the manuscript's title and its actual content, as well as the issues noted above, I recommend a major revision. The authors should:
- revise the title to reflect the current scope more accurately,
- significantly expand the manuscript to include original or comparative data on bioactive compounds specifically from Andean plant varieties, with clear discussion of their therapeutic potential and effective extraction methods.
Reviewer Comment:
The title of the manuscript—“Andean plants as a source of bioactive molecules for modern therapeutic approaches”—suggests a specific focus on bioactive compounds derived from Andean plant species and their potential therapeutic applications. However, the content of the article does not sufficiently support this focus. Only limited attention is given to the actual properties of bioactive molecules specifically from Andean varieties of quinoa, amaranth, and lupin.
In particular:
- The majority of the information presented is general and qualitative, often repeating well-known data about bioactive compounds found in these crops, with little to no focus on their Andean origin. For instance in sections 2 and 3, the authors describe quinoa, amaranth, and lupin as historically important crops with origins in the Andes, and they discuss the influence of the Andean environment on the phytochemical properties of these plants. However, the article includes very little—if any—research specifically focused on these plant species grown in the Andean region.
- No quantitative data or comparative analysis is provided regarding bioactive profiles of Andean versus non-Andean plant varieties.
- The manuscript does not include any original studies or in-depth discussion of therapeutic effects, despite this being explicitly mentioned in the title. Potential therapeutic effects are not the same as actual therapeutic effects.
- The repetition of well-known information across multiple sections further weakens the overall impact. For example, the information presented in lines 599 to 605 consists of general chemical facts that are not specific to the extraction of compounds from quinoa, amaranth, or lupin. In this section, I would expect not only a qualitative description but also a quantitative comparison of the most effective methods for the extraction of phenolic compounds from these plants, especially in relation to Andean varieties.
- In section 5.2 author describe to general about antioxidants. For the
- In Sections 4 and 6, the authors describe specific groups of bioactive compounds found in quinoa and amaranth, including their chemical structures and health-related effects. However, this content is repeated in Sections 5 and 7, resulting in unnecessary redundancy. To improve clarity and coherence, I recommend consolidating and restructuring these sections.
In contrast, the section on lupin is the most clearly structured and informative part of the manuscript. It provides a more focused and coherent overview, which could serve as a model for improving the presentation of content in the other sections.
- Figures are not referenced or contextualized in the main text, reducing their contribution to the overall narrative.
- Authors should provide a clear description of the methodology used to select and review the literature, including databases searched, keywords, and inclusion/exclusion criteria.
- Furthermore, while the manuscript seems to aim for a comprehensive review, it lacks the critical depth and synthesis expected of such an article. The text often reiterates generally known information
Given the significant gap between the manuscript's title and its actual content, as well as the issues noted above, I recommend a major revision. The authors should:
- revise the title to reflect the current scope more accurately,
- significantly expand the manuscript to include original or comparative data on bioactive compounds specifically from Andean plant varieties, with clear discussion of their therapeutic potential and effective extraction methods.
Author Response
- The majority of the information presented is general and qualitative, often repeating well-known data about bioactive compounds found in these crops, with little to no focus on their Andean origin. For instance in sections 2 and 3, the authors describe quinoa, amaranth, and lupin as historically important crops with origins in the Andes, and they discuss the influence of the Andean environment on the phytochemical properties of these plants. However, the article includes very little—if any—research specifically focused on these plant species grown in the Andean region.
- Response: We thank the reviewer for this observation. The term "Andean plants" in the manuscript refers to species that originated in the Andes and that continue to be widely cultivated in this region. While quinoa, amaranth, and lupin are now grown in diverse agroecological zones across the world, our primary focus is on varieties that are native to the Andes. To enhance this clarity, we have revised the manuscript and incorporated additional references to specific Andean varieties and their cultivation context.
- No quantitative data or comparative analysis is provided regarding bioactive profiles of Andean versus non-Andean plant varieties.
- Response: We thank the reviewer for this observation. However, we respectfully clarify that the main objective of this review is not to conduct a comparative analysis between Andean and non-Andean varieties, but rather to provide an integrative synthesis of the bioactive compounds and associated health benefits of edible plants originated at the Andean region. Our emphasis lies in describing the biochemical richness and therapeutic relevance of these ancestral crops based on current scientific evidence.
- The manuscript does not include any original studies or in-depth discussion of therapeutic effects, despite this being explicitly mentioned in the title. Potential therapeutic effects are not the same as actual therapeutic effects.
Response: Although we appreciate the reviewer’s comments, we disagree with this one. As the title of our paper also explicitly mentioned, this is a comprehensive review, so it would not be expected to present original data. Moreover, our in-depth review of numerous in vitro, in vivo, and clinical studies that report measurable physiological outcomes—including anti-inflammatory, antidiabetic, hypocholesterolemic, and hepatoprotective effects—linked to specific bioactive compounds from quinoa, amaranth, and lupin, allowed us to discuss in an objective and critical way their actual therapeutic effects (sections 4 through 10). Furthermore, Table 4, also created after in-depth analysis, summarizes results only from clinical trials, providing therapeutic benefits rather than merely potential uses.
- The repetition of well-known information across multiple sections further weakens the overall impact. For example, the information presented in lines 599 to 605 consists of general chemical facts that are not specific to the extraction of compounds from quinoa, amaranth, or lupin. In this section, I would expect not only a qualitative description but also a quantitative comparison of the most effective methods for the extraction of phenolic compounds from these plants, especially in relation to Andean varieties.
- Response: The aim of this review is not to describe extraction methods or compare Andean versus non-Andean varieties, but rather to synthesize recent peer-reviewed literature on plants that are of Andean origin, focusing on their phytochemical diversity and health-related bioactivities. We believe this perspective provides value by emphasizing the unique biochemical adaptations shaped by Andean environmental conditions.
- In section 5.2 author describe to general about antioxidants.
Response: We have rewritten this section to focus specifically on antioxidant compounds found in Andean cultivars of quinoa, such as quercetin, vanillic acid, and rutin, and their reported antioxidant activities (e.g., DPPH scavenging capacity).
- In Sections 4 and 6, the authors describe specific groups of bioactive compounds found in quinoa and amaranth, including their chemical structures and health-related effects. However, this content is repeated in Sections 5 and 7, resulting in unnecessary redundancy. To improve clarity and coherence, I recommend consolidating and restructuring these sections.
Response: We thank the reviewer for this valuable observation. Rather than fully restructuring the sections, we carefully reviewed them in detail to identify and eliminate redundancies between the descriptions of bioactive compounds and their health effects. Repetitive content was removed to ensure clarity and avoid duplication, while preserving all relevant scientific information.
In contrast, the section on lupin is the most clearly structured and informative part of the manuscript. It provides a more focused and coherent overview, which could serve as a model for improving the presentation of content in the other sections.
- Figures are not referenced or contextualized in the main text, reducing their contribution to the overall narrative.
Response: All figures (1–4) are now referenced in the appropriate sections of the main text, and explanatory details originally placed in figure legends have been integrated into the body text to enhance context and readability.
- Authors should provide a clear description of the methodology used to select and review the literature, including databases searched, keywords, and inclusion/exclusion criteria.
- Furthermore, while the manuscript seems to aim for a comprehensive review, it lacks the critical depth and synthesis expected of such an article. The text often reiterates generally known information.
We appreciate the reviewer’s suggestion. While this manuscript is a narrative review rather than a systematic or bibliometric analysis, we acknowledge the value of clarifying the literature selection process. We have therefore included a brief note in the Introduction mentioning that we conducted a comprehensive literature search on bioactive compounds from Chenopodium quinoa, Amaranthus spp., and Lupinus spp., including their biological activities and health benefits, using the scientific databases Web of Science (WoS) and Scopus. We selected peer-reviewed original research articles, review papers, conference proceedings, and book chapters published in English or Spanish within the last ten years. Exceptionally, we included a limited number of older publications when they provided fundamental information on the botanical or chemical characteristics of the studied plants . However, we did not apply systematic review protocols such as PRISMA, as our objective was to provide an integrative, qualitative synthesis rather than a meta-analysis.
Given the significant gap between the manuscript's title and its actual content, as well as the issues noted above, I recommend a major revision. The authors should:
- revise the title to reflect the current scope more accurately,
- significantly expand the manuscript to include original or comparative data on bioactive compounds specifically from Andean plant varieties, with clear discussion of their therapeutic potential and effective extraction methods.
Response: Thank you for your valuable suggestion. In response, we have revised the title to: “Recent Advances in the Therapeutic Potential of Bioactive Molecules from Andean-Origin Plants.” We believe this updated title more accurately reflects the scope and focus of the manuscript, emphasizing both the therapeutic dimension and the origin of the plant species discussed.
Reviewer 2 Report
Comments and Suggestions for Authors
The manuscript entitled “Andean Plants as a Source of Bioactive Molecules for Modern Therapeutic Approaches: A Comprehensive Review” provides valuable knowledge on the bioactive compounds and health benefits of quinoa, amaranth, and lupin. The paper is very interesting, but some comments are below.
The title should be more precise because the authors base their research on only three species, which grew in different regions of the world. Additionally, there should be no period at the end of the title.
The discussed plants should be mentioned in the keywords.
Line 53- 57: In their introductions, the authors describe how environmental conditions in the Andean region have driven the evolution of plants, resulting in the accumulation of diverse secondary metabolites with improved bioactivities. However, the text does not explicitly state where the plant varieties in question come from.
The spaces between paragraphs/lines are too large because they artificially increase the volume of the manuscript.
Section 3 appears to be unnecessary or should be supplemented with information on environmental factors affecting phytochemicals, considering other regions of the world where these plants are cultivated. In keeping with the theme, the information in the manuscript should focus solely on plants growing in the Andes.
The information in the Bioactive Ingredients and Health Applications sections should perhaps be combined, as there is often duplication of content. Repetitive content should be simplified for clarity.
In many places in the text, there is no information about what part of the plant was examined.
Figures 1-4 were not cited in the text, furthermore, the information contained under the titles should also be placed in the text and not in the title.
Line 474:”DPPH and ABTS” – correct to DPPH* and ABTS+*
Line 567: „…Amaranth” - add Latin name
Line 585:"ml" correct to "mL"
Line 594: (Manyelo et al., 2020) – quote correctly
Subsection 6.1 should contain more information regarding peptides, their sequences, and other information based on literature sources.
Line 635:”[91] S” – missing a dot
Subsection 7.1 – add antifungal activity to the title
Line 712: add amaranth variety
Line 720-721: this information should be in section 7.2
Line 769: „sample (db.)” – what does “db” mean.
Line 773: “ABTS+ assays” - plus is unnecessary
Section 7.3 - The content should be arranged according to the order in the title
Line 897: This information should be added to the peptides subsection
Line 903: This information should be added to the phenolic compounds subsection
Section 8 – The order of discussing the components should be maintained similarly to the other cases
Line 968: This information should be added to the additional health benefits
Line 1166-169: “Interestingly, L. albus landraces exhibited the highest alkaloid concentrations, with major compounds including lupanine, 13α-hydroxylupanine, and albine….L. angustifolius contained lupanine, 13α-hydroxylupanine, and angustifoline” -connect these sentences.
Line 1281: “ with an ABTS value” – correct it
Line 1313: “/g DPPH” - check if this is correct
Section 9.3.- The content should be arranged according to the order in the title
Line 1372: This information should be added to the peptides subsection
The title of Table 3 should be above the table. In Table 3. change the name “against bacterial infection” to antibacterial or a more correct one.
Section 10. The title “Potencial use of the bioactive molecules from Andean plants based on clinical trials” - should be revised, as this section focuses on the consumption of quinoa, amaranth, or lupin and their products.
Line 1591-1599: “NO3- or NO2-“ – 3- and 2- should be superscript
“caused mild pro-oxidant activity” – are you sure about that
A valuable addition would be including information on the levels of antioxidant enzyme activity in plants discussed in this paper (such as catalase, peroxidase, etc.).
Furthermore, the manuscript does not address the potential risk related to the antinutritional factors of the plants discussed.
Author Response
The manuscript entitled “Andean Plants as a Source of Bioactive Molecules for Modern Therapeutic Approaches: A Comprehensive Review” provides valuable knowledge on the bioactive compounds and health benefits of quinoa, amaranth, and lupin. The paper is very interesting, but some comments are below.
The title should be more precise because the authors base their research on only three species, which grew in different regions of the world. Additionally, there should be no period at the end of the title.
Response: Thank you for your suggestion regarding the title.
We agree that the original title could be more precise given the scope of the manuscript. Accordingly, we have revised the title to:
“Recent Advances in the Therapeutic Potential of Bioactive Molecules from Plants of Andean Origin”.
We believe this version more accurately reflects the focus on Chenopodium quinoa, Amaranthus spp., and Lupinus spp., and clarifies that the review addresses species with Andean origins, regardless of their current global distribution.
We have also removed the period at the end of the title, in accordance with journal guidelines.
The discussed plants should be mentioned in the keywords.
Response: Keywords have been updated as suggested.
Line 53- 57: In their introductions, the authors describe how environmental conditions in the Andean region have driven the evolution of plants, resulting in the accumulation of diverse secondary metabolites with improved bioactivities. However, the text does not explicitly state where the plant varieties in question come from.
Response: Thank you for your observation. We have updated the references in the introduction and explicitly clarified the geographic origin of the quinoa, amaranth, and lupin varieties discussed. The revised text now specifies that the reviewed species are native to the Andean region and highlights how their adaptation to high-altitude environments has influenced their phytochemical profiles.
The spaces between paragraphs/lines are too large because they artificially increase the volume of the manuscript.
Response: Formatting issues have been addressed to ensure consistent and appropriate spacing throughout the document.
Section 3 appears to be unnecessary or should be supplemented with information on environmental factors affecting phytochemicals, considering other regions of the world where these plants are cultivated. In keeping with the theme, the information in the manuscript should focus solely on plants growing in the Andes.
Response:Thank you for your comment. We would like to clarify that the focus of this review is on quinoa, amaranth, and lupin as species of Andean origin, rather than exclusively on specimens currently cultivated in the Andean region. Accordingly, we have included in section 3.3 available data comparing the phytochemical profiles of these plants when grown under different environmental conditions, which helps to illustrate how factors such as altitude, soil composition, and UV exposure influence their bioactive compound content.
The information in the Bioactive Ingredients and Health Applications sections should perhaps be combined, as there is often duplication of content. Repetitive content should be simplified for clarity.
Response: Thank you for this observation. We carefully reviewed the sections on Bioactive Ingredients and Health Applications and have removed repetitive content to enhance clarity and avoid redundancy. However, we have chosen to keep the sections separate in order to maintain a logical distinction between the chemical nature of the compounds and the specific health effects they exert, which we believe adds structural clarity for the reader.
In many places in the text, there is no information about what part of the plant was examined.
Response: We have revised the text to indicate the specific plant part examined (e.g., seeds, leaves, sprouts) in all relevant cases.
Figures 1-4 were not cited in the text, furthermore, the information contained under the titles should also be placed in the text and not in the title.
Response: Figures have now been cited in the main text, and their descriptive content has been moved into the corresponding paragraphs to improve contextual integration.
Line 474:”DPPH and ABTS” – correct to DPPH* and ABTS+*
We thank you for the observation. The revision was made as suggested.
Line 567: „…Amaranth” - add Latin name
We thank you for the observation. The revision was made as suggested.
Line 585:"ml" correct to "mL"
Line 594: (Manyelo et al., 2020) – quote correctly
We thank you for the observation. The revision was made as suggested.
Subsection 6.1 should contain more information regarding peptides, their sequences, and other information based on literature sources.
Response: Thank you for your suggestion. In response, we have expanded the section on Amaranth Peptides to include detailed information on peptide sequences, their molecular weights, physicochemical properties
Line 635:”[91] S” – missing a dot
We thank you for the observation. The revision was made as suggested.
Subsection 7.1 – add antifungal activity to the title
Thank you for the suggestion. We would like to clarify that we used the term antimicrobial in the subsection title because it encompasses activity against a broad range of microorganisms, including fungi
Line 712: add amaranth variety
We thank you for the observation. The revision was made as suggested.
Line 720-721: this information should be in section 7.2
Response: Thank you for your suggestion. We have corrected this section.
Line 769: „sample (db.)” – what does “db” mean.
Response: Thank you for your suggestion. We have corrected this word.
Line 773: “ABTS+ assays” - plus is unnecessary
Section 7.3 - The content should be arranged according to the order in the title
Response: Thank you for your suggestion. We have corrected this section.
Line 897: This information should be added to the peptides subsection
Response: Thank you for your suggestion. We have corrected this section.
Line 903: This information should be added to the phenolic compounds subsection
Response: Thank you for your suggestion. We have corrected this section.
Section 8 – The order of discussing the components should be maintained similarly to the other cases
Response: Thank you for your suggestion. We have corrected the order of the sections.
Line 968: This information should be added to the additional health benefits
Response: Thank you for your suggestion. We have corrected this section.
Line 1166-169: “Interestingly, L. albus landraces exhibited the highest alkaloid concentrations, with major compounds including lupanine, 13α-hydroxylupanine, and albine….L. angustifolius contained lupanine, 13α-hydroxylupanine, and angustifoline” -connect these sentences.
Response: Thank you for your suggestion. We have corrected this sentence.
Line 1281: “ with an ABTS value” – correct i
Response: Thank you for your suggestion. We have corrected this section.
Line 1313: “/g DPPH” - check if this is correct
Response: Thank you for your suggestion. It’s correct.
Section 9.3.- The content should be arranged according to the order in the title
Response: Thank you for your suggestion. We have corrected this section.
Line 1372: This information should be added to the peptides subsection
Response: Thank you for your suggestion. We have corrected this section.
The title of Table 3 should be above the table. In Table 3. change the name “against bacterial infection” to antibacterial or a more correct one.
Response: Thank you for your suggestion. We have corrected this section.
Section 10. The title “Potencial use of the bioactive molecules from Andean plants based on clinical trials” - should be revised, as this section focuses on the consumption of quinoa, amaranth, or lupin and their products.
Response: Thank you for reaching this important point that might create some confusion. However, as it was explained in lines 1613-17: “... there is an important number of trials already completed or been conducted with most of the active ingredients deeply described in the previous sections, it is almost impossible to extrapolate those to the consumption of the whole product. For this reason, in this section we analized the few clinical trials acomplishing with that premise, i.e. use the whole product”. Therefore, and per your suggestion, this section is now entitled “Potencial therapeutic use of the Andean plants based on clinical trials”.
Line 1591-1599: “NO3- or NO2-“ – 3- and 2- should be superscript
Thank you for your suggestion. We have corrected this mistake.
A valuable addition would be including information on the levels of antioxidant enzyme activity in plants discussed in this paper (such as catalase, peroxidase, etc.).
Response: We sincerely thank the reviewer for their thoughtful suggestion to include information on the levels of antioxidant enzyme activity, such as catalase and peroxidase, in the Andean plants discussed. We fully recognize the importance of these enzymes in the plants’ response to oxidative stress and their role in shaping biochemical adaptations to environmental challenges.
However, the primary focus of our manuscript is to review the role of secondary metabolites (e.g., phenolics, peptides, polysaccharides) in quinoa, amaranth, and lupin, with an emphasis on their direct effects on human health and potential applications in nutrition and medicine. While antioxidant enzymes are undoubtedly relevant to plant physiology, discussing their activity levels in these plants falls outside the scope of this review, which is centered on bioactive compounds with therapeutic and nutritional relevance.
To maintain alignment with our objectives, we believe that including a detailed discussion on enzyme activities might divert attention from the core focus on secondary metabolites and their health benefits. Nonetheless, we greatly appreciate the reviewer’s suggestion, as it highlights an interesting avenue for future research.
Furthermore, the manuscript does not address the potential risk related to the antinutritional factors of the plants discussed.
Response: We deeply appreciate the reviewer’s comment, which underscores the importance of addressing antinutritional factors in the Andean plants covered in our manuscript. We acknowledge that compounds such as saponins, phytates, and alkaloids, present in quinoa, amaranth, and lupin, may pose challenges to digestibility and food safety, and discussing these factors is essential for providing a balanced perspective on their therapeutic and nutritional potential.
In our manuscript, we chose to focus on bioactive compounds with health benefits, as our primary goal was to highlight their value in functional foods and pharmacological applications. However, we fully agree that omitting a discussion on antinutritional factors could limit the comprehensive understanding of these plants, particularly in the context of their direct consumption or integration into modern diets.
To address this concern, in the modified version of the manuscript we include a new subsection (“Considerations on Antinutritional Factors.”) in Section 11 ("Concluding Remarks").
Reviewer 3 Report
Comments and Suggestions for Authors
The review paper contains a great deal of collected information on the manuscript topic. It is written correctly but needs improvement. I have included all comments for the authors in the attached file.

The English could be improved to more clearly express the research.
Author Response
Why were these plants chosen for this review paper?
We deeply appreciate the reviewer’s comment,we have added to the abstract the following "Lupinus, Amaranthus, and Chenopodium species were selected for this review due to their Andean origin, long-standing role in traditional diets, and growing scientific interest in their unique phytochemical profiles and therapeutic potential.”
Please add a subsection on terpenoids. fatty acids
Thank you for your suggestion. While we recognize the relevance of terpenoids and fatty acids in the phytochemical composition of Andean plants, we have chosen to focus this review on polyphenols, polysaccharides, proteins, and peptides, in alignment with the scope and thematic focus of the special issue (Plant-Derived Bioactive Polyphenols, Polysaccharides, Proteins, and Peptides, and Their Health Benefits). We hope this clarifies our decision to limit the discussion to the categories outlined in the issue’s objectives.
It should be written in italics.
All mistakes were corrected
The title of the chapter should be corrected. e.g. Health applications of quinoa bioactive compounds
All tittles observations were corrected as suggested.
The title of the table should be placed above the table.
The title was placed above the table
Reviewer 4 Report
Comments and Suggestions for Authors
Title:
Andean Plants a Source of Bioactive Molecules for Modern Therapeutic Approaches. A comprehensive review.
The manuscript by Barba-Ostria C. et al. is a review collecting and discussing data on bioactive compounds present in Andean plants and their health applications, focusing on 3 key species, i.e., quinoa (Chenopodium quinoa), amaranth (Amaranthus spp.) and lupine (Lupinus spp.).
Evidence from in vitro, preclinical and clinical studies was included.
The collected information was organized and detailed in the introduction and 9 other sections, plus concluding remarks, and also in 4 figures and 3 tables, all supported by a body of evidence of 238 references, of which a significant number are recent publications.
Although this thorough review provides valuable information, there is still room for improvement of the manuscript to ensure its uniqueness and scientific soundness.
General comments:
- Recommendation to check abbreviations throughout the manuscript and reduce their number by eliminating unnecessary ones.
- Recommendation to check all scientific names and rewrite them correctly, with italics only when necessary.
- Compliance with the requirements in the instructions for authors, for tables and figures.
- Checking the content of tables and figures.
- Minor editing of English language required.
Specific comments:
Title:
- I suggest including a special character, such as "-" or ":", after Andean Plants
Andean Plants - a Source of Bioactive Molecules for Modern Therapeutic Approaches: A comprehensive review
Abstract:
line 22 - Amaranthus spp. - spp. without italics
Introduction and Body of text:
For such a comprehensive review, it should include a Methodology section after the Introduction that:
- presents the selection criteria
(how the filtering took place: how many years of studies and types of studies, including in vitro, in vivo and clinical studies)
- and specifies the databases used (such as PubMed, Scopus, MEDLINE, Cochrane and ScienceDirect etc.).
Please be consistent with all abbreviations in the manuscript, including their meanings the first time they are mentioned.
- e.g., see line 614 - ACE was already abbreviated earlier on line 529
- same for line 967...
- LPH - was first used in the manuscript without explanation of the abbreviation (line 956) and was abbreviated only when used later (line 1338)
- LDL - was first used in the manuscript without explanation of the abbreviation (line 535)
- was abbreviated only when used later (line 895)
- and further on, in the manuscript, the same abbreviation was not always kept: e.g. at line 1460 - using LDL-C.
Please try to reduce the use of rarely used abbreviations:
for example,
- BD, BDdbs, CQP, FD, etc. - these terms appear only once in the manuscript and the abbreviation is not justified
- line 736 - the abbreviation ATCE is not justified (it is used only here)
Here are just a few examples.
Please review all abbreviations and their frequency of use.
Use italics when necessary (e.g. scientific names in the body of the text and in the references section) and remove inappropriate italics:
- line 220 - Haplopappus - should be italicized
- line 263 - in Chenopodium quinoa: Willd - should be written without italics
- same – at line 440
- line 332 – Quinoa – without italics
- line 631 - hybridus L. - L. should not be italicized
- line 658 - tricolor L. - L. should not be italicized
- line 698 - spp. without italics
- line 852 - Amaranthus caudatus - L. should not be italicized
- line 1162 - Lupinus - spp. without italics
Also, there is no uniformity regarding the term "species", which is used after the genus names:
- sometimes abbreviated as: "spp.",
- other times not abbreviated: "species".
line 946 - please correct "Lupius" to "Lupinus"
Figure 2
As the first figure in the manuscript in which certain abbreviations appear, the abbreviations should be explained in the legend.
Table 1:
- has no legend.
There are special characters and abbreviations in Table 1 that should be explained in the legend.
- "containing" - without italics
Please consider reviewing all legends (all tables and figures).
Concluding remarks:
These are consistent with the review's findings and have been summarized accordingly.
References:
I recommend replacing or removing very old references wherever possible.
For example,
ref. 21 - year of publication 1981
ref. 150 - year of publication 1992
ref. 151 - year of publication 1998
ref. 152 - year of publication 1967
ref. 155 - year of publication 1986
ref. 218 - year of publication 1981
ref. 219 - year of publication 1992
ref. 220 - year of publication 1994
Final conclusions:
In my opinion, the authors have produced a comprehensive work of interest to the journal's readers.
Overall, the review is a bit long, with the data for the 3 species being approached in a complex manner.
Some future prospects were mentioned, but not the strengths and limitations of the study.
Therefore, I suggest major revisions to improve its quality before publication.
Author Response
- I suggest including a special character, such as "-" or ":", after Andean Plants
Andean Plants - a Source of Bioactive Molecules for Modern Therapeutic Approaches: A comprehensive review
We deeply appreciate the reviewer’s comment. The title was changed as suggested by the reviewers
Abstract:
line 22 - Amaranthus spp. - spp. without italics
Response: Thank you for your suggestion. We have corrected this mistake.
Introduction and Body of text:
For such a comprehensive review, it should include a Methodology section after the Introduction that:
- presents the selection criteria
(how the filtering took place: how many years of studies and types of studies, including in vitro, in vivo and clinical studies)
- and specifies the databases used (such as PubMed, Scopus, MEDLINE, Cochrane and ScienceDirect etc.).
We appreciate the reviewer’s suggestion. While this manuscript is a narrative review rather than a systematic or bibliometric analysis, we acknowledge the value of clarifying the literature selection process. We have therefore included a brief note in the Introduction mentioning that we conducted a comprehensive literature search on bioactive compounds from Chenopodium quinoa, Amaranthus spp., and Lupinus spp., including their biological activities and health benefits, using the scientific databases Web of Science (WoS) and Scopus. We selected peer-reviewed original research articles, review papers, conference proceedings, and book chapters published in English or Spanish within the last ten years. Exceptionally, we included a limited number of older publications when they provided fundamental information on the botanical or chemical characteristics of the studied plants . However, we did not apply systematic review protocols such as PRISMA, as our objective was to provide an integrative, qualitative synthesis rather than a meta-analysis.
Please be consistent with all abbreviations in the manuscript, including their meanings the first time they are mentioned.
- e.g., see line 614 - ACE was already abbreviated earlier on line 529
- same for line 967...
- LPH - was first used in the manuscript without explanation of the abbreviation (line 956) and was abbreviated only when used later (line 1338)
- LDL - was first used in the manuscript without explanation of the abbreviation (line 535)
- was abbreviated only when used later (line 895)
- and further on, in the manuscript, the same abbreviation was not always kept: e.g. at line 1460 - using LDL-C.
Please try to reduce the use of rarely used abbreviations:
for example,
- BD, BDdbs, CQP, FD, etc. - these terms appear only once in the manuscript and the abbreviation is not justified
- line 736 - the abbreviation ATCE is not justified (it is used only here)
Here are just a few examples.
Thank you for your suggestion. We agreed and the unnecessary abbreviations were eliminated.
Please review all abbreviations and their frequency of use.
Use italics when necessary (e.g. scientific names in the body of the text and in the references section) and remove inappropriate italics:
Thank you for your observation. We have revised the manuscript and reference list to ensure that scientific names are italicized appropriately and that any incorrect use of italics has been removed.
line 220 - Haplopappus - should be italicized
- line 263 - in Chenopodium quinoa: Willd - should be written without italics
- same – at line 440
- line 332 – Quinoa – without italics
- line 631 - hybridus L. - L. should not be italicized
- line 658 - tricolor L. - L. should not be italicized
- line 698 - spp. without italics
- line 852 - Amaranthus caudatus - L. should not be italicized
- line 1162 - Lupinus - spp. without italics
Response: Thank you for your suggestion. We have corrected these mistakes.
Also, there is no uniformity regarding the term "species", which is used after the genus names:
- sometimes abbreviated as: "spp.",
- other times not abbreviated: "species".
line 946 - please correct "Lupius" to "Lupinus"
Thank you for pointing this out. We have revised the manuscript to ensure consistent use of the term species following genus names, in accordance with taxonomic conventions.
Figure 2
As the first figure in the manuscript in which certain abbreviations appear, the abbreviations should be explained in the legend.
Response: Thank you for your suggestion. We have corrected these mistakes.
Table 1:
- has no legend.
There are special characters and abbreviations in Table 1 that should be explained in the legend.
Please consider reviewing all legends (all tables and figures).
Thank you for your comment. We have updated the legend of Table 1 to include explanations for all special characters and abbreviations used, ensuring clarity and consistency for the reader.
Concluding remarks:
These are consistent with the review's findings and have been summarized accordingly.
References:
I recommend replacing or removing very old references wherever possible.
For example,
ref. 21 - year of publication 1981
ref. 150 - year of publication 1992
ref. 151 - year of publication 1998
ref. 152 - year of publication 1967
ref. 155 - year of publication 1986
ref. 218 - year of publication 1981
ref. 219 - year of publication 1992
ref. 220 - year of publication 1994
Thank you for your recommendation. We have replaced older references with more recent and relevant sources wherever possible. However, in cases where historical context or foundational background was necessary, we retained key older references to preserve the integrity and continuity of the information presented.
Final conclusions:
In my opinion, the authors have produced a comprehensive work of interest to the journal's readers.
Overall, the review is a bit long, with the data for the 3 species being approached in a complex manner.
Some future prospects were mentioned, but not the strengths and limitations of the study.
Therefore, I suggest major revisions to improve its quality before publication.
Round 2
Reviewer 2 Report
Comments and Suggestions for Authors
-
Reviewer 4 Report
Comments and Suggestions for Authors
Dear authors,
The revised version of your manuscript is now much improved compared to the initial version.
I consider this version suitable for publication.